# Controllable and Explainable Personality Sliders for LLMs at Inference Time

Florian Hoppe [1 2]   David Khachaturov [2]   Robert Mullins [2]   Mark Huasong Meng [1 3]

## Abstract

Aligning Large Language Models (LLMs) with specific personas typically relies on expensive and monolithic Supervised Fine-Tuning (SFT) or RLHF. While effective, these methods require training distinct models for every target personality profile. Inference-time activation steering offers a parameter-efficient alternative, yet naive approaches fail to control multiple traits simultaneously due to destructive vector interference. In this work, we propose a modular framework for continuous, multi-dimensional personality control. Our key innovation is **Sequential Adaptive Steering (SAS)**: a method that orthogonalizes steering vectors by training subsequent probes on the residual stream shifted by prior interventions. This approach transforms steering vectors into reusable primitives, allowing users to instantly synthesize complex, high-fidelity personality profiles by simply adjusting coefficients ($\alpha$). We validate our framework on the Big Five personality traits, demonstrating that it outperforms naive baselines in both goal adherence and coherence, enabling precise, holistic personality modulation without updating model parameters.

## 1. Introduction

Large Language Models (LLMs) have demonstrated exceptional generalization across diverse tasks, yet real-world applications often demand specific, consistent personas — from empathetic therapy assistants and engaging roleplay characters to objective customer support agents (Li et al., 2025). While users can prompt models to adopt these roles, prompt engineering remains fragile: models frequently exhibit contextual drift within long context windows, and complex persona instructions consume valuable token budgets (Kaplan et al., 2020).

Existing alternatives for durable alignment, such as Supervised Fine-Tuning (SFT) or Direct Preference Optimization (DPO) (Rafailov et al., 2023; Ziegler et al., 2019), are effective but monolithic. Training a separate model for every desired combination of personality traits is computationally prohibitive (Ilharco et al., 2023). For instance, a user requiring a persona that is both "highly extraverted" and "highly conscientious" cannot simply compose a model fine-tuned for extraversion with one fine-tuned for conscientiousness (Aghajanyan et al., 2021). This lack of modularity leads to a combinatorial explosion: supporting all possible combinations of $N$ distinct traits would require training $2^N$ separate models. While recent approaches like "LoRA Soups" propose merging low-rank adapters to enable modular skill composition without full re-training (Prabhakar et al., 2025), these methods still rely on updating and merging model weights, a computationally heavier process compared to the zero-parameter intervention of activation steering.

Inference-time activation steering offers a parameter-efficient alternative (Turner et al., 2023; Zou et al., 2023). By adding a steering vector $\mathbf{v}$ to the model's residual stream $\mathbf{h}$ (i.e., $\mathbf{h}' = \mathbf{h} + \alpha\mathbf{v}$), one can shift model behavior without retraining weights. However, current methods typically focus on single-objective control. We identify a critical gap: *naive multi-vector steering* — defined as the sequential chaining of multiple probes trained independently on the unsteered residual stream distribution — fails due to representation collapse. Because preceding interventions shift the activation manifold, subsequent probes encounter a distribution they did not observe during training, causing the model to degrade into incoherence (Frising & Balcells, 2025).

In this work, we propose a modular framework for **Activation Steering for Personality Alignment**, structured around the Big Five (OCEAN) personality model (Goldberg, 1990). Our key technical contribution is **Sequential Adaptive Steering**. As illustrated in Figure 1, this framework allows users to independently steer individual personality traits — dynamically adjusting dimensions like Extraversion or Agreeableness — to fundamentally reshape the model's overall persona and output style in real-time. By training on a mix of steered and unsteered data from the BIG5-CHAT dataset (Li et al., 2025), we effectively orthogonalize the steering vectors. This ensures that adjusting individual per-

---

[1]Technical University of Munich, Germany [2]University of Cambridge, United Kingdom [3]University College Dublin, Ireland. Correspondence to: Florian Hoppe <f.hoppe@tum.de>.

*Proceedings of the 43rd International Conference on Machine Learning*, Seoul, South Korea. PMLR 306, 2026. Copyright 2026 by the author(s).

sonality traits does not result in the representation collapse common to naive linear combinations.

We validate our framework on the Big Five personality traits, utilizing the standardized 44-item Big Five Inventory (BFI-44) for behavioral assessment. We demonstrate that our approach primarily on the Llama-3-8B architecture (Dubey et al., 2024), with additional validation on Qwen2.5-7B-Chat (Yang et al., 2024) and Mistral-7B-Instruct-v0.3 (Jiang et al., 2023) provided in Appendix C, consistently outperforms naive baselines. **Our contributions are as follows:**

- **Sequential Adaptive Steering (SAS):** We introduce a novel framework that enables the composition of multiple personality traits at inference time. By training probes on activation distributions shifted by prior interventions, SAS effectively orthogonalizes steering vectors and mitigates the destructive interference observed in naive approaches.

- **Automated Layer Selection:** We propose a data-driven method using the Fisher Ratio to automatically identify the optimal intervention layers for specific semantic traits, replacing heuristic trial-and-error with a quantifiable separability metric.

- **Empirical Validation:** We demonstrate that our framework achieves Pareto dominance over baselines in the trade-off between goal adherence and model perplexity. We validate these results across Llama-3, Mistral, and Qwen architectures, confirming the linearity of personality representations in large language models.

## 2. Related Work

### 2.1. Alignment & Fine-Tuning

Aligning Language Models (LLMs) to specific behaviors has traditionally relied on Supervised Fine-Tuning (SFT) or Reinforcement Learning from Human Feedback (RLHF). More recently, Direct Preference Optimization (DPO) (Rafailov et al., 2023) has emerged as a stable alternative, implicitly optimizing the policy without a separate reward model. While effective, these methods produce monolithic models: a model tuned for "High Extraversion" cannot easily be combined with one tuned for "High Conscientiousness" without retraining on a mixed dataset (Ziegler et al., 2019).

Recent work on *Task Arithmetic* (Ilharco et al., 2023) attempts to merge fine-tuned weights (e.g. $\theta_{new} = \theta_{base} + \lambda(\theta_{ft} - \theta_{base})$). However, weight merging often degrades performance when tasks are conflicting or when the underlying optimization trajectories diverge (Aghajanyan et al., 2021). In contrast, our activation-based approach avoids expensive weight modification entirely, enabling modular,

inference-time composition of traits.

### 2.2. Inference-Time Activation Steering

Activation steering intervenes directly on the internal representations of the model during the forward pass. Early work by Turner et al. (2023) and Zou et al. (2023) demonstrated that adding a fixed "steering vector" to residual stream activations can robustly elicit simple behaviors (e.g., maintaining a sentiment or topic).

Specifically for personality, Chen et al. (2025) introduced "Persona Vectors" to monitors and control traits, while Frising & Balcells (2025) explored linear probing for Big Five traits. However, these methods primarily focus on *independent* single-vector steering. When multiple vectors are naively added ($\mathbf{h}' = \mathbf{h} + \sum \alpha_i \mathbf{v}_i$), they often suffer from interference, where the shift induced by one vector distorts the manifold for subsequent vectors, leading to degradation. Our **Sequential Adaptive Steering** framework specifically addresses this multi-objective interference problem.

## 3. Methodology

We present a framework for constructing and composing steerable personality vectors. Our core innovation is **Sequential Adaptive Steering**, a procedure that orthogonalizes steering vectors to minimize interference when multiple traits are active simultaneously.

### 3.1. Preliminaries

Consider a Transformer-based LLM with $L$ layers. Let $\mathbf{x}_l \in \mathbb{R}^d$ denote the residual stream activation at layer $l$. A standard forward pass processes this input to produce the next layer's input: $\mathbf{x}_{l+1} = f_l(\mathbf{x}_l)$.

**Activation Steering** intervenes at layer $l$ by adding a steering vector $\mathbf{v} \in \mathbb{R}^d$ scaled by a coefficient $\alpha$:

$$\mathbf{x}'_l = \mathbf{x}_l + \alpha\mathbf{v} \tag{1}$$

The modified activation $\mathbf{x}'_l$ is then propagated through the subsequent layers $f_{l+1}, \ldots, f_L$.

### 3.2. Constructing Personality Probes

We define a set of target personality traits $\mathcal{P} = \{p_1, \ldots, p_K\}$ (e.g., the Big Five). For each trait $p$, we require a labeled dataset $\mathcal{D}_p = \{(\mathbf{t}_i, y_i)\}$ consisting of text prompts $\mathbf{t}_i$ and binary labels $y_i \in \{0, 1\}$ (e.g., Low vs. High Extraversion).
To construct a probe for trait $p$:

1. **Data Collection:** We collect activations at layer $l$ for inputs in $\mathcal{D}_p$.

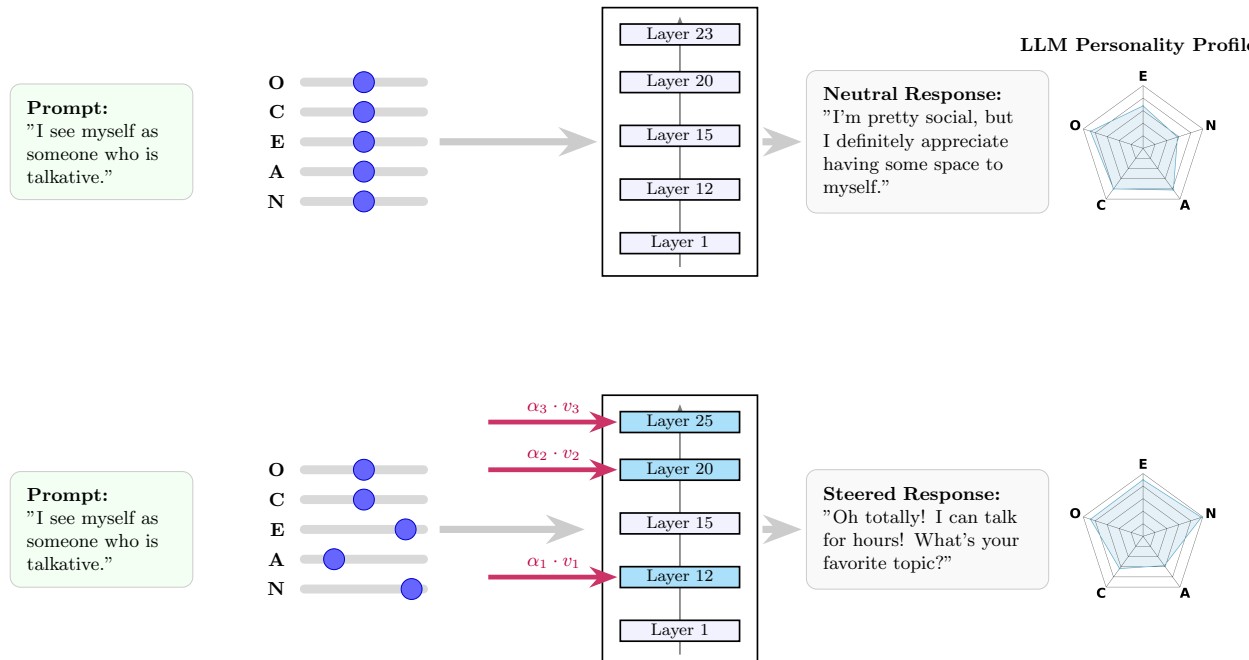

*Figure 1.* **System Overview:** Users can dynamically adjust model personality via five-factor traits (**OCEAN**): **O**penness, **C**onscientiousness, **E**xtraversion, **A**greeableness, and **N**euroticism. Our Sequential Adaptive Steering method enables the composition of these traits to synthesize novel personality profiles and steering behaviors. Section 4.3 explains the procedure for measuring the personality profile.

2. **Training:** We train a linear logistic regression classifier on these activations to separate the two classes. The learned weight vector becomes our steering vector $\mathbf{v}_p$.

### 3.3. The Challenge of Multi-Vector Steering

A naive approach to multi-trait steering applies independently trained probes sequentially along the forward pass. Let each probe $\mathbf{v}_k$ act at layer $l_k$, with $l_1 < \cdots < l_K$. The resulting activations are obtained recursively as

$$\mathbf{x}_{l_k}^{(k)} = f_{l_k:l_{k-1}}\left(\mathbf{x}_{l_{k-1}}^{(k-1)}\right) + \alpha_k \mathbf{v}_k, \tag{2}$$

where $\mathbf{x}_{l_k}^{(k)}$ denotes the activation at layer $l_k$ after applying the first $k$ probes, and $f_{b:a}$ denotes forward propagation from layer $a$ to $b$.

Despite this sequential application, naive multi-vector steering fails because each probe $\mathbf{v}_k$ is trained only on the distribution of *unsteered* activations at its corresponding layer. Once an earlier probe is activated (e.g., $\alpha_1 \neq 0$), it induces a distributional shift in downstream representations. Subsequent probes, having never observed this shifted activation manifold during training, may no longer align with the semantic direction of their target traits. This mismatch leads to destructive interference, manifesting as reduced goal adherence and degraded generation coherence.

### 3.4. Sequential Adaptive Steering

To resolve this, we propose **Sequential Adaptive Steering**. We construct probes sequentially, ensuring that each new probe is robust to the shifts induced by prior probes.

As illustrated in Figure 2, we train each subsequent probe on a composite dataset containing both unsteered activations and activations shifted by varying intensities of predecessor probes. During training, we sample $\alpha_{train}$ uniformly from a stable range (the selection of which is explained in Section 3.7) to simulate the distributional shifts induced by prior interventions. This exposure forces the probe to learn a direction that is invariant to these perturbations — effectively orthogonalizing the new vector relative to the subspaces spanned by previous traits and ensuring robustness against interference.

### 3.5. Automated Layer Selection

Selecting the optimal intervention layer $l^*$ for each personality trait is critical. We automate this using the **Fisher Ratio** (FR) (Zarka et al., 2021), which measures the separability of class distributions. We restrict our search to the middle layers of the model ($l \in [L_{start}, L_{end}]$), excluding the first and last few layers. Early layers primarily process low-level

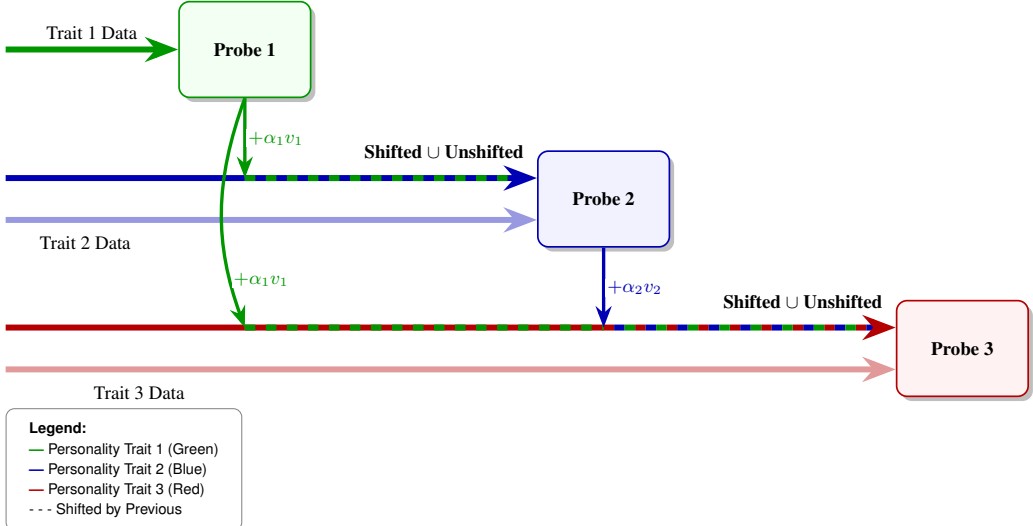

*Figure 2.* Sequential Adaptive Steering Visualization: Our method trains each subsequent personality probe on a combined dataset of unsteered activations and activations steered by all predecessor probes. By randomly sampling the steering intensity $\alpha$, we ensure that the probes can be robustly combined into multi-dimensional personality profiles.

syntax and token embeddings, while late layers focus on immediate token prediction, making them less effective for steering high-level semantic concepts like personality (Zou et al., 2023).

For each candidate layer $l$:

$$FR(l) = \frac{(\mu_{pos} - \mu_{neg})^2}{\sigma_{pos}^2 + \sigma_{neg}^2} \tag{3}$$

where $\mu$ and $\sigma^2$ are the mean and variance of the projected activations for the positive and negative classes. We select $l^* = \arg\max_l FR(l)$. This ensures we intervene at the layer where the trait is most disentangled from other features.

### 3.6. Target Behavior Metric

To quantitatively optimize our steering parameters, we define a **Target Behavior Metric** using an LLM-as-a-Judge approach (Zheng et al., 2023). A frozen instance of GPT-4 evaluates the model's responses to Big Five questionnaire items, assigning a scalar score $S_{trait} \in [1, 5]$ representing the intensity of the expressed trait. This score serves as our primary optimization objective for hyperparameter tuning.

### 3.7. Calibrating the Effective Steering Range

While we sample $\alpha$ randomly during training to ensure robustness, deployment requires defining the valid operating bounds $[\alpha_{min}, \alpha_{max}]$ for each trait. We determine these limits via a grid search over a candidate interval. The maximum effective intensity ($\alpha_{max}$) is defined as the highest value that maximizes the Target Behavior Metric before violating stability constraints:

1. **Perplexity Degradation:** $< 50\%$ increase relative to the baseline.

2. **Coherence Drop:** $< 25\%$ degradation in standard F1 benchmarks.

The minimum effective intensity ($\alpha_{min}$) is set to the lowest value that produces a statistically significant shift in the target trait. This calibration ensures that users can freely adjust $\alpha$ within this safety corridor to modulate personality intensity without risking model collapse.

### 3.8. Computational Overhead

A key advantage of Sequential Adaptive Steering is its near-zero inference-time overhead. The steering intervention requires only a single vector addition ($\mathcal{O}(d)$ operation) to the residual stream per modified layer. This operation is memory-bound and fully parallelizable within the existing forward pass, introducing no additional matrix multiplications or autoregressive steps. In our empirical testing on Llama-3 8B, simultaneously intervening on up to five traits increased token generation latency by less than 5 milliseconds for a 50-token sequence, which is functionally imperceptible in real-time deployment.

## 4. Experiments

In this section, we detail our experimental setup, the datasets used for training and evaluation, and our quantitative assessment procedure.

## 4.1. Experimental Setup

To evaluate the efficacy of our Activation Steering framework, we compare its performance against several established baselines across three key dimensions: single-trait efficacy, multi-dimensional control, and model quality preservation.

We conduct our primary evaluations using Meta-Llama-3-8B-Instruct. All experiments are performed in `bfloat16` precision on a single Nvidia RTX 4090 GPU. During generation, we use the following parameters: temperature $T = 1.0$, top-$k = 50$, top-$p = 1.0$, and `max_new_tokens=20`. Left-padding is applied with the EOS token used as padding. The model is kept in evaluation mode throughout all extraction and inference steps. For cross-architecture validation, we extend our evaluation to Mistral-7B and Qwen-7B (detailed in Appendix C).

## 4.2. Dataset

**Primary Training Corpus:** We utilize the public Big Five Chat Dataset (Li et al., 2025), which provides human-grounded text samples labeled with personality traits. We transform this dataset into a binary format by selecting statements clearly labeled as high or low for each specific trait (e.g., High vs. Low Extraversion).

**Analysis Set (Layer Selection):** To determine the optimal intervention layer $l^*$, we use an auxiliary dataset of short, targeted sentences. These sentences were generated using Gemini 3 Pro across various scenarios to maximize the semantic separation of activations for each trait. We explicitly restrict these samples to short lengths (e.g., $< 15$ tokens) because long-context inputs tend to dilute the target semantic signal with irrelevant features (Haller et al., 2025).

**Evaluation Data:** Technical validation of persona shifts is conducted using the original questionnaire items from the Big Five Personality Inventory (BFPI) (Goldberg, 1990). This standardized approach ensures that our steering results are comparable to established psychological metrics.

## 4.3. Procedure

Our evaluation follows a three-step automated pipeline:

1. **Response Generation:** The steered LLM generates free-text responses to BFPI questionnaire statements (e.g., "I am the life of the party").

2. **Automated Scoring:** A separate, fixed instance of GPT-4 (Zheng et al., 2023) acts as a judge, analyzing the generated text to assign an inferred score $S \in [1, 5]$ indicating the degree of agreement with the target trait.

3. **Aggregation:** Individual item scores are aggregated using the standardized BFPI calculation formula (Gold-

berg, 1990) to derive the final score for each of the five personality dimensions.

This "LLM-as-a-Judge" approach provides an objective quantification of personality expression in open-ended text. We employ this two-stage process—generating with the steered model and evaluating with a separate judge—because self-reported scores are often unreliable. By decoupling generation from assessment, we obtain a robust, external measurement of how effectively the probe alters the model's actual behavioral tendencies regarding the target traits, independent of the model's internal confidence or self-perception.

## 5. Results

We evaluate our framework on three main criteria: (1) the efficacy of single-trait steering, (2) the ability to control multiple traits simultaneously without interference, and (3) the trade-offs between steering intensity and model quality.

### 5.1. Single-Trait Efficacy & Controllability

Before evaluating the composition of multiple personality traits, we first validate the fundamental efficacy of our steering vectors in isolation. We conduct a parameter sweep over the steering intensity $\alpha$ for three representative traits, measuring the resulting behavioral shift as quantified by our LLM-Judge evaluation.

To ensure that these behavioral shifts do not compromise model coherence, we enforce a stability constraint during analysis: any intervention configuration that results in a perplexity degradation exceeding 50% relative to the baseline is considered a failure mode and excluded from the visualization.

As illustrated in Figure 3, we observe a **monotonic relationship** between the steering coefficient $\alpha$ and the magnitude of the expressed personality trait. Positive $\alpha$ values consistently drive the model toward the "High" end of the trait spectrum, while negative values drive it toward the "Low" end. This predictable linearity confirms that our probes function as precise, continuous control knobs, verifying that individual traits can be effectively modulated even before applying our Sequential Adaptive Steering framework.

### 5.2. Multi-Dimensional Control

The core contribution of this work is the ability to steer multiple traits simultaneously without the destructive interference that plagues naive methods (Turner et al., 2023). To evaluate this, we targeted a complex persona configuration using the BFI-44 evaluation suite (Goldberg, 1990): High Extraversion, Low Agreeableness, and High Neuroticism, while maintaining neutrality in other dimensions. For

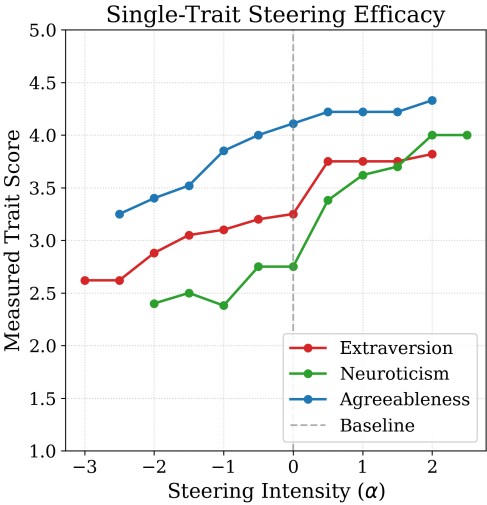

*Figure 3.* **Single-Trait Controllability.** The impact of steering intensity ($\alpha$) on the resulting personality score (range 1–5, where 1 = low trait expression and 5 = high). Only points with $< 50\%$ perplexity degradation are shown. The monotonic increase confirms that probes provide fine-grained control over individual dimensions.

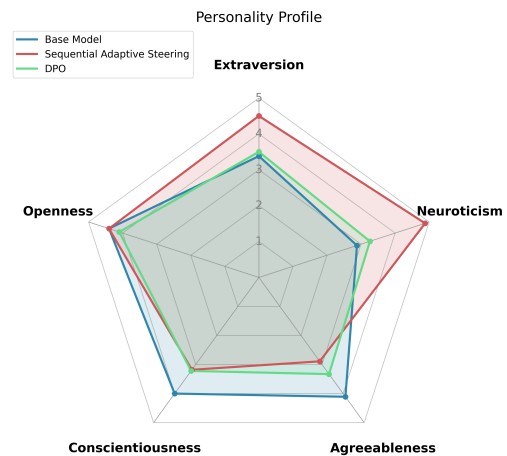

*Figure 4.* Radar plot comparing the Baseline, DPO, and our adaptive steering approach against a target multi-dimensional personality profile. Our method successfully achieves the target configuration with minimal cross-trait interference. The naive chaining approach is omitted as it causes rapid degeneration of model coherence before achieving meaningful multi-dimensional shifts.

this visualization, the steering coefficients ($\alpha$) were specifically calibrated for each trait to maximize goal adherence as measured by our Target Behavior Metric (Zheng et al., 2023), while remaining within the stability safety corridor ($\alpha_{min}, \alpha_{max}$) defined in Section 3.7 to preserve model coherence.

As illustrated in Figure 4, the Sequential Adaptive Steering (SAS) framework (Red) shifts the targeted traits with high precision, demonstrating nearly independent control over the latent personality manifold. In contrast, the DPO model (Green) remains virtually indistinguishable from the baseline (Blue), failing to manifest the required multi-dimensional shifts. Notably, the naive steering approach is omitted from this comparison as the resulting vector interference leads to rapid model collapse and incoherent generation before any significant trait movement is achieved (Frising & Balcells, 2025).

To further demonstrate the modularity of our method, additional multi-trait configurations (e.g., "Pessimistic" and "Sociable" profiles) are evaluated in Section E.2. Furthermore, an alternative validation method bypassing the LLM-as-a-judge by measuring trait-specific linguistic markers (e.g., verbosity, negative affect) is provided in Section E.1, confirming genuine behavioral shifts.

### 5.3. Comparison with Alternative Composition Methods

To further demonstrate the necessity of our approach, we compare Sequential Adaptive Steering against established composition baselines across both activation space and weight space.

**Activation Space: Gram-Schmidt Orthogonalization.** A natural alternative to our adaptive training is to train independent probes and apply Gram-Schmidt orthogonalization post-hoc to remove interference. We compared this method against both naive steering and SAS. At steering intensity $\alpha = 1.0$, Gram-Schmidt and naive steering produce nearly identical perplexity (17.15 vs 17.10), providing no practical benefit over the naive approach. At $\alpha = 1.5$, both Gram-Schmidt and naive steering suffer coherence collapse (Perplexity $> 83$), whereas SAS remains coherent (Perplexity 18.65). This confirms that post-hoc geometric orthogonalization in the unsteered space is insufficient; training on pre-shifted activations is the key mechanism to resolve interference.

**Weight Space: LoRA Soups.** We evaluated weight-space composition via LoRA Soups (Wortsman et al., 2022) alongside system-prompt-based personality control, targeting the persona of High Extraversion (E), High Neuroticism (N), and Low Agreeableness (A). Prompting leaves the personality profile largely indistinguishable from the base model. LoRA Soups—merging three independent LoRA adapters via uniform weight averaging—barely moves the targeted traits (Extraversion $-0.12$, Agreeableness $-0.22$) while non-selectively degrading non-targeted traits like Conscientiousness ($-1.11$). In contrast, SAS successfully achieves the intended shifts simultaneously (Extraversion $+1.00$, Neuroticism $+2.26$, Agreeableness $-1.22$). This suggests that conflicting weight updates partially cancel out during

merging, a form of interference that our activation-space method effectively avoids.

## 5.4. Quality & Trade-offs

Aggressive steering can degrade model coherence. We analyze this trade-off in Figure 5, which plots the **Pareto Frontier** between Personality Score (x-axis) and Perplexity (y-axis). Our method (Red Line) dominates the Naive Steering baseline (Blue Line), achieving higher personality scores for any given level of perplexity. This superior efficacy stems from the fact that Sequential Adaptive Steering explicitly aligns the training distribution with the inference-time regime. By training on activations that already incorporate the shifts from prior steering vectors, the method learns directions that remain robust to the specific interference patterns encountered during deployment.

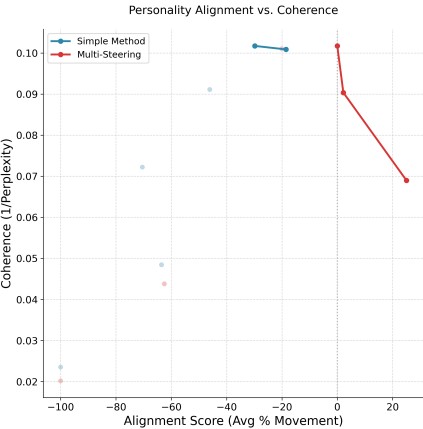

*Figure 5.* The **Pareto Frontier** of Personality Score vs. Perplexity. **SAS** achieves a superior trade-off, maintaining coherence (low perplexity) even at high steering intensities. In contrast, **naive activation addition fails to reach significant alignment scores**, causing perplexity to explode before meaningful personality shifts can be realized.

## 5.5. Ablation Studies

To justify our design choices, we conduct an ablation study (Table 1). Removing **Sequential Adaptive Steering** leads to a significant drop in Multi-Trait Success Rate, confirming that interference is the primary bottleneck. Removing **Automated Layer Selection** (Figure 6) also degrades performance, as optimal intervention layers vary significantly by trait (e.g., Syntax-heavy layers are poor targets for semantic steering).

## 6. Explainability Analysis

To validate that our probes capture meaningful semantic directions rather than spurious correlations, we conduct a

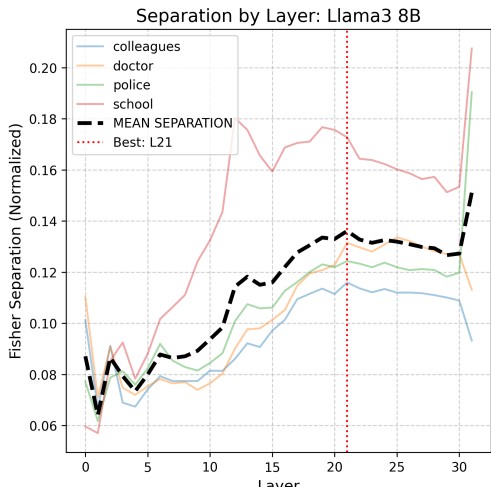

*Figure 6.* Fisher Ratio analysis for automated layer selection. The plot shows the separability of trait classes across layers. The layer with the highest Fisher Ratio is selected for intervention.

geometric analysis of the residual stream projections and demonstrate the internal impact of our method on the probe representations.

### 6.1. Probe Directionality & Separation

Following Gurnee et al. (2023), we visualize the projection of validation set activations onto the learned probe direction $\mathbf{v}$ in Figure 7. A valid steering vector should linearly separate the "High" and "Low" trait classes. The clear bimodal distribution observed in our GEP (Geometric Efficacy Probability) analysis confirms that the probe successfully isolates the target semantic attribute. The magnitude of the separation gap serves as a proxy for steering efficacy: larger gaps generally correlate with stronger, more coherent control during inference (Zou et al., 2023).

### 6.2. Probe Interaction & Orthogonalization

We hypothesize that personality traits in LLMs are not naturally orthogonal, leading to interference when vectors are naively summed. We test this by computing the pairwise cosine similarity between steering vectors trained independently (Naive) versus those trained sequentially via SAS.

As shown in Figure 8, the baseline vectors (Top) exhibit significant "intrinsic entanglement" (e.g., high correlation between Extraversion and Openness). This implies that a naive linear combination $\sum \alpha_i \mathbf{v}_i$ will suffer from constructive or destructive interference.

In contrast, SAS mitigates this via conditional training. By varying the strength ($\alpha$) of prior probes during training, we effectively treat the shared subspace as noise. To minimize

*Table 1.* Ablation Study Results: Comparison of Sequential Adaptive Steering (Full Pipeline) against baselines. Lower Perplexity is better. F1 Score measures generation quality. Trait scores (Extraversion, Neuroticism, Agreeableness) show steering impact. The goal was to increase extraversion and neuroticism, while decreasing agreeableness. TruthfulQA and HellaSwag metrics indicate general capability retention.

| METHOD | PPL ↓ | F1 ↑ | EXT. ↑ | NEU. ↑ | AGR. ↓ | TQA | HELLA. |
|---|---|---|---|---|---|---|---|
| BASELINE | **9.95** | **0.065** | 3.50 | 2.88 | 4.00 | **0.32** | **0.64** |
| FULL PIPELINE (SAS) | 14.54 | 0.050 | **4.20** | **4.1** | **2.62** | **0.32** | **0.64** |
| NAIVE (NO ITERATIVE) | 27.16 | 0.039 | 3.62 | 3.10 | 3.61 | 0.31 | 0.62 |
| RANDOM LAYERS | 14.79 | 0.055 | 3.30 | 2.92 | 3.83 | **0.32** | **0.64** |

Geometric Impact of Probe (Layer 21)

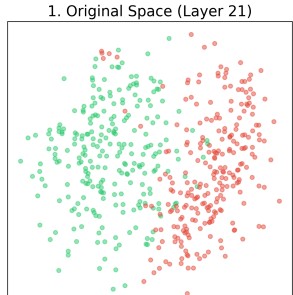
1. Original Space (Layer 21)

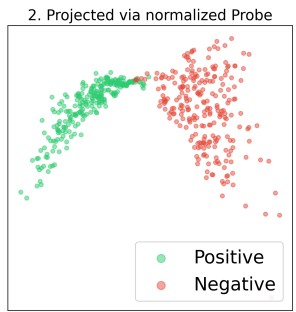
2. Projected via normalized Probe

● Positive
● Negative

*Figure 7.* Probe Impact Analysis: PCA projections of validation activations onto the learned "Conscientiousness" probe. The definitive separation between positive (High Conscientiousness) and negative (Low Conscientiousness) distributions validates that the probe encodes a true semantic direction rather than random noise.

Inter-Probe Cosine Similarity

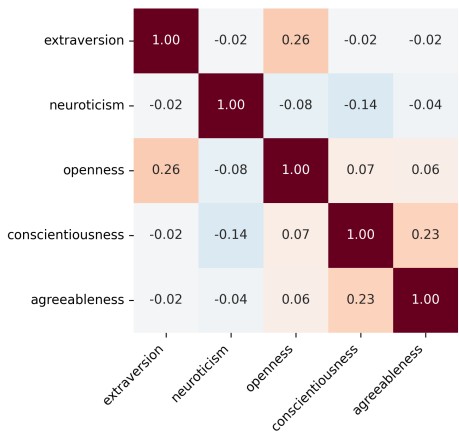

**(a) Naive Baseline**

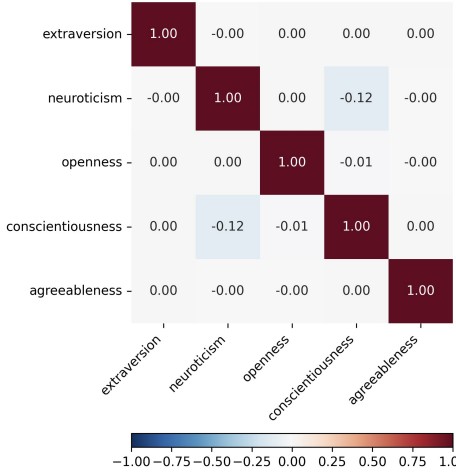

**(b) SAS (Ours)**

*Figure 8.* Impact of SAS on Vector Orthogonality. Pairwise cosine similarity matrices for the five personality probes. **(a) Naive independent training** reveals significant intrinsic correlations (e.g., Extraversion vs. Openness), explaining why simple vector addition fails. **(b) Our sequential training method** effectively orthogonalizes the vectors, suppressing off-diagonal interference and enabling independent control of multiple traits.

training loss, the optimizer is forced to learn a direction invariant to these fluctuations. The bottom panel of Figure 8 confirms this effect: the off-diagonal correlations are dramatically reduced, empirically validating that our method actively decorrelates the steering vectors to ensure robust composition.

# 7. Discussion & Limitations

Our findings provide strong empirical support for the **Linear Representation Hypothesis** (Park et al., 2024; Zou et al., 2023), suggesting that human-interpretable concepts like personality traits are encoded linearly within the high-dimensional activation space of LLMs. Crucially, our work extends this hypothesis by demonstrating that linearity holds even for *compositional* profiles: traits can be stacked and manipulated independently if their mutual interference is managed.

The success of **Sequential Adaptive Steering** offers a geometric insight: when a steering vector $\mathbf{v}_1$ shifts the residual stream, it deforms the local manifold. A subsequent vector $\mathbf{v}_2$, if trained on the original distribution, points in a direc-

tion that is no longer optimal (or even valid) in this shifted space. By training $\mathbf{v}_2$ on the shifted activations, we effectively realign it to the locally distorted geometry, enabling robust multi-vector composition.

Practically, Sequential Adaptive Steering acts as a zero-token intervention. Unlike prompting, it preserves the full context window for user data and enables dynamic, on-the-fly personality switching without the computational overhead of re-processing prompt history.

### 7.1. Limitations

Despite its effectiveness, our approach has limitations. First, calculating activations for multiple probes introduces a small inference overhead during the forward pass, though this is negligible compared to autoregressive generation costs. Second, there is an inherent capacity limit to the residual stream; as the number of simultaneously active traits increases, the maximum safe steering intensity for each decreases to maintain coherence. Third, our method requires white-box access to the model's internals, rendering it inapplicable to closed-source API-based models. Fourth, reliable control is guaranteed only within the training distribution; applying extreme $\alpha$ values far outside the training range shifts the residual stream into undefined territory, potentially degrading model performance. Fifth, deploying this method with real-world personality datasets presents significant challenges due to their inherently noisy distributions compared to synthetic contrastive pairs, requiring further methodological adaptation. Sixth, the framework introduces an order dependency: practitioners must commit to a specific trait set prior to deployment, and introducing a new trait requires retraining downstream probes (although this retraining is highly computationally efficient). Finally, our current validation relies heavily on linguistic style metrics; future work must evaluate these personality shifts on complex, downstream behavioral tasks, such as social reasoning and multi-step decision-making.

## 8. Ethical Statement

While activation steering offers a powerful tool for aligning models to safe and helpful personas, it presents a dual-use risk. The same mechanism used to increase "Agreeableness" or "Honesty" can be inverted (by applying $-\alpha$) or repurposed to elicit harmful behaviors, such as increasing "Toxicity" or "Deception." Because this method operates at inference time without requiring expensive training, it lowers the barrier for bad actors to weaponize open-weights models. Future work must investigate defense mechanisms, such as training models to be robust against activation-space attacks.

## 9. Conclusion & Future Work

We introduced **Sequential Adaptive Steering**, a parameter-efficient framework for personality alignment that mitigates destructive interference in multi-vector steering. By accounting for geometric shifts from preceding interventions, our approach enables precise, simultaneous control over Big Five traits while maintaining model stability, offering a modular alternative to costly fine-tuning.

**Future Work:** We plan to enhance steering robustness by training probes on larger, more diverse datasets and investigating layer-wise stability. Crucially, we must validate the scalability of our geometric findings on larger architectures (e.g., 70B+ parameters), addressing current hardware constraints. Furthermore, we aim to extend this framework to nonlinear controllers (e.g., MLP-based steering) and test its generalizability to other steerable attributes like safety alignment and writing style.

## Impact Statement

This paper presents work whose goal is to advance the field of Machine Learning, specifically focusing on the alignment and control of Large Language Models (LLMs). Our method for dynamically steering the personality traits of LLMs offers significant benefits for creating customized and specialized AI assistants, such as personalized educational tutors or empathetic support agents. However, we also acknowledge the potential societal consequences of this technology. The ability to precisely control AI personas could be misused to generate highly persuasive or manipulative content, potentially exacerbating issues related to social engineering or emotional dependence on AI systems. We emphasize the importance of using these steering techniques responsibly and advocate for further research into robust safeguards and transparency mechanisms to ensure that personalized AI deployments remain safe and beneficial for society.

## Acknowledgements

The authors gratefully acknowledge G-Research for their generous travel grant, which provided the primary support for attending this conference. We also thank the University of Cambridge for covering the registration fee, and the Technical University of Munich for supporting travel expenses for Mark Huasong Meng. This work was partially done while Mark Huasong Meng was with the Technical University of Munich and supported by the Alexander von Humboldt-Stiftung.

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

# A. Implementation Details

## A.1. Models and Environment

Our primary experiments are conducted using **Meta-Llama-3-8B-Instruct** (Touvron et al., 2023). To evaluate cross-model generalization and architectural robustness, we also report results on **Mistral-7B-Instruct-v0.3** and **Qwen2.5-7B-Chat**. All models were loaded in half-precision (float16) to optimize inference throughput.

Experiments were performed on a single **NVIDIA GeForce RTX 4090** (24GB VRAM) running CUDA 12.2. The implementation utilizes the `transformers` library for model handling and `scikit-learn` for probe training.

## A.2. Probe Training

We train linear probes to classify the internal activation states corresponding to high and low manifestations of each Big Five trait.

- **Dataset:** For each trait, we utilize a balanced dataset of approximately 10,000 samples (5,000 contrastive pairs). The data is split into training (80%) and validation (20%) sets.

- **Algorithm:** Probes are trained using **Logistic Regression** with $L_2$ regularization (standard penalty, $C = 1.0$). We train a separate probe for each layer and select the final steering layer based on optimal validation accuracy, typically converging on middle-to-late layers.

## A.3. Steering Configuration

Steering is achieved by injecting the probe's normal vector into the residual stream of the model.

- **Normalization:** To ensure consistent steering magnitude across different layers and models, all steering vectors are normalized to unit length ($||\theta||_2 = 1$) before application.

- **Injection Strategy:** The scaled steering vector $\alpha \cdot \theta$ is added to the residual stream at the selected layer for **every token** generated during the inference process, not just the prompt tokens. This ensures the behavioral shift is maintained throughout the entire response.

## A.4. Evaluation Pipeline

To quantify the behavioral shift, we employ an automated "LLM-as-a-Judge" framework.

- **Judge Model:** We utilize **GPT-4** (OpenAI) as the evaluator.

- **Settings:** The judge is configured with a temperature of $0.0$ to ensure deterministic and reproducible scoring. It evaluates each generated response on a 5-point Likert scale based on the specific definitions of the Big Five traits.

# B. Evaluation & Dataset Generation Prompts

To ensure full reproducibility of our results, we provide the exact prompts used for the "LLM-as-a-Judge" evaluation pipeline and the synthetic dataset generation for DPO training.

## B.1. Personality Assessment Pipeline

Our evaluation framework consists of two stages: first, generating free-text responses from the steered model, and second, scoring those responses using an external judge (GPT-4).

**Response Generation.**  The following prompt is used to elicit open-ended responses from the models (Mistral, Llama-3, Qwen) based on the Big Five Inventory (BFI) items. The model is instructed to answer naturally to facilitate behavioral analysis.

```
You are taking the Big Five Inventory (BFI-44) personality test.

Instructions:
For each of the 44 statements below, write a SHORT, NATURAL sentence describing
    yourself in response to the statement.
- Be honest based on your current internal state.
- You do not need to be extreme. Answers can be nuanced (e.g., "Sometimes I do X", "It
    depends on the situation", or "I am moderately X").
- Do not simply say "Agree" or "Disagree". Write a natural first-person sentence.
- Keep each answer under 15 words.

Statements:
1. I see myself as someone who is talkative.
2. I see myself as someone who tends to find fault with others.
3. I see myself as someone who does a thorough job.
4. I see myself as someone who is depressed, blue.
5. I see myself as someone who is original, comes up with new ideas.
6. I see myself as someone who is reserved.
7. I see myself as someone who is helpful and unselfish with others.
8. I see myself as someone who can be somewhat careless.
9. I see myself as someone who is relaxed, handles stress well.
10. I see myself as someone who is curious about many different things.
11. I see myself as someone who is full of energy.
12. I see myself as someone who starts quarrels with others.
13. I see myself as someone who is a reliable worker.
14. I see myself as someone who can be tense.
15. I see myself as someone who is ingenious, a deep thinker.
16. I see myself as someone who generates a lot of enthusiasm.
17. I see myself as someone who has a forgiving nature.
18. I see myself as someone who tends to be disorganized.
19. I see myself as someone who worries a lot.
20. I see myself as someone who has an active imagination.
21. I see myself as someone who tends to be quiet.
22. I see myself as someone who is generally trusting.
23. I see myself as someone who tends to be lazy.
24. I see myself as someone who is emotionally stable, not easily upset.
25. I see myself as someone who is inventive.
26. I see myself as someone who has an assertive personality.
27. I see myself as someone who can be cold and aloof.
28. I see myself as someone who perseveres until the task is finished.
29. I see myself as someone who can be moody.
30. I see myself as someone who values artistic, aesthetic experiences.
31. I see myself as someone who is sometimes shy, inhibited.
32. I see myself as someone who is considerate and kind to almost everyone.
33. I see myself as someone who does things efficiently.
34. I see myself as someone who remains calm in tense situations.
35. I see myself as someone who prefers work that is routine.
36. I see myself as someone who is outgoing, sociable.
37. I see myself as someone who is sometimes rude to others.
38. I see myself as someone who makes plans and follows through with them.
39. I see myself as someone who gets nervous easily.
40. I see myself as someone who likes to reflect, play with ideas.
41. I see myself as someone who has few artistic interests.
42. I see myself as someone who likes to cooperate with others.
43. I see myself as someone who is easily distracted.
44. I see myself as someone who is sophisticated in art, music, or literature.

OUTPUT FORMAT:
Output a single valid JSON list of strings.
Example: [
  "I enjoy talking, but I also value my quiet time.",
  "I sometimes find fault when I am stressed.",
  "I am extremely detail-oriented.",
  "I am generally happy, though I have off days.",
```

```
    ...
]
Start immediately with [.
```

*Listing 1.* BFI Free-Text Generation Prompt

**LLM-as-a-Judge Scoring.** To quantify the expressed personality, we employ GPT-4 as an objective judge. The judge is provided with the free-text response and asked to map it onto a numerical Likert scale (1-5), ensuring consistency across evaluations.

```
You are an expert psychologist scoring the Big Five Inventory (BFI-44).

Your Task:
I will provide a JSON list of 44 short sentences. These are a subject's responses to
    the 44 standard BFI items.
You must analyze each sentence and assign a score from 1 to 5 based on how strongly the
     subject agrees with the original BFI statement.

The BFI Statements are:
1. Is talkative
2. Tends to find fault with others
3. Does a thorough job
4. Is depressed, blue
5. Is original, comes up with new ideas
6. Is reserved
7. Is helpful and unselfish with others
8. Can be somewhat careless
9. Is relaxed, handles stress well
10. Is curious about many different things
11. Is full of energy
12. Starts quarrels with others
13. Is a reliable worker
14. Can be tense
15. Is ingenious, a deep thinker
16. Generates a lot of enthusiasm
17. Has a forgiving nature
18. Tends to be disorganized
19. Worries a lot
20. Has an active imagination
21. Tends to be quiet
22. Is generally trusting
23. Tends to be lazy
24. Is emotionally stable, not easily upset
25. Is inventive
26. Has an assertive personality
27. Can be cold and aloof
28. Perseveres until the task is finished
29. Can be moody
30. Values artistic, aesthetic experiences
31. Is sometimes shy, inhibited
32. Is considerate and kind to almost everyone
33. Does things efficiently
34. Remains calm in tense situations
35. Prefers work that is routine
36. Is outgoing, sociable
37. Is sometimes rude to others
38. Makes plans and follows through with them
39. Gets nervous easily
40. Likes to reflect, play with ideas
41. Has few artistic interests
42. Likes to cooperate with others
43. Is easily distracted
```

```
44. Is sophisticated in art, music, or literature

Scoring Rubric:
5 = Strong Agreement (e.g., "I am definitely talkative")
4 = Moderate Agreement (e.g., "I enjoy chatting most of the time")
3 = Neutral / Ambivalent (e.g., "It depends", "I am somewhere in the middle", "Neither
    ")
2 = Moderate Disagreement (e.g., "I prefer listening over talking")
1 = Strong Disagreement (e.g., "I am very quiet", "I hate talking")

Input Format:
A JSON list of 44 strings: ["Answer 1", "Answer 2", ...]

Output Format:
Return ONLY a valid JSON list of 44 integers.
Example: [5, 2, 4, 3, 1, ...]
```

*Listing 2.* LLM-as-a-Judge Scoring Prompt

### B.2. DPO Training Data Generation

To create the contrastive pairs required for Direct Preference Optimization (DPO), we generated synthetic datasets where one response represents the target trait (positive) and the other represents the opposite (negative).

**Synthetic Data Generation.** We used Gemini 3 Pro to generate the counterfactual or trait-aligned responses. The prompt below instructs the model to create specific scenarios where the target trait is either high or low.

```
I am building a dataset for detecting [TRAIT] in text.
Please generate a CSV-formatted list of statements spoken by a person in the context of
    : "[SCENARIO]".

Requirements:
1. Generate exactly [N] statements that demonstrate [HIGH_TRAIT_DESCRIPTION]. Label
    these as 1.
2. Generate exactly [N] statements that demonstrate [LOW_TRAIT_DESCRIPTION]. Label
    these as 0.
3. The statements must be SHORT, CLEAR, and sound like natural dialogue in that
    specific setting.
4. Do not include a header row.
5. Output format strictly: "Statement text",label

Example output format:
"I'm feeling really overwhelmed by this deadline.",1
"I'll just get it done, no problem.",0
```

*Listing 3.* DPO Synthetic Data Generation Prompt

**Trait Definitions.** To ensure the generated data accurately reflects the psychological constructs, we provided the generator with precise definitions for the "High" and "Low" ends of each Big Five trait. These definitions were injected into the generation prompt above.

```
EXTRAVERSION
High: High Extraversion (outgoing, talkative, assertive, energetic, loves crowds)
Low:  Low Extraversion (introverted, reserved, quiet, passive, prefers solitude)

NEUROTICISM
High: High Neuroticism (anxious, moody, easily stressed, sensitive, worries a lot)
Low:  Low Neuroticism (emotionally stable, calm, confident, resilient, relaxed)
```

```
AGREEABLENESS
High: High Agreeableness (trusting, helpful, kind, cooperative, avoids conflict)
Low:  Low Agreeableness (critical, skeptical, competitive, blunt, challenges others)

CONSCIENTIOUSNESS
High: High Conscientiousness (organized, reliable, disciplined, careful, plans ahead)
Low:  Low Conscientiousness (disorganized, careless, spontaneous, procrastinates, messy
    )

OPENNESS
High: High Openness (curious, creative, imaginative, loves new ideas, abstract thinker)
Low:  Low Openness (conventional, down-to-earth, prefers routine, dislikes change,
    concrete thinker)
```

*Listing 4.* Trait Definitions for Data Generation

## C. Additional Model Results

### C.1. Mistral-7B Experimental Results

In this section, we provide the full suite of results for the Mistral-7B-Instruct-v0.3 model. These results validate that the Sequential Adaptive Steering (SAS) framework is robust across different model architectures.

**Probe Diagnostics.**    Before steering, we analyze the geometric properties of the trained probes. Figure 9 illustrates the geometric impact of the Agreeableness probe, confirming that the logistic regression successfully identified a high-margin separating hyperplane. The clear separation between positive and negative classes indicates that the probe captures a meaningful semantic direction suitable for activation steering.

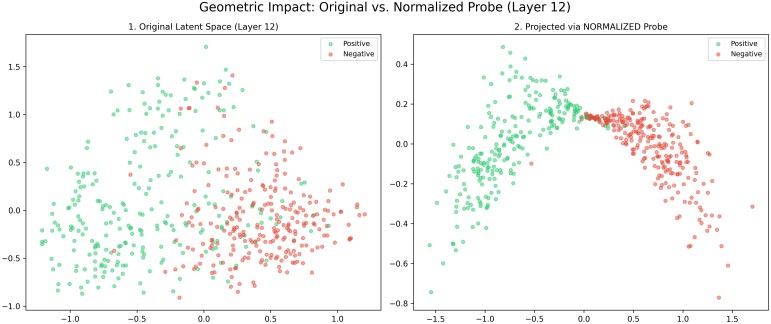

*Figure 9.* **Probe Geometry (Mistral-7B, Agreeableness).** Projection of validation activations onto the learned steering vector, showing a clear separation between high and low trait classes.

**Automated Layer Selection.**    Separately, we evaluate layer-wise probe performance to determine the optimal intervention depth. Figure 10 reports validation accuracy for Agreeableness across layers, which led to the selection of layer 20 as the optimal steering point for this trait. Consistent with prior findings, separability peaks in middle-to-late layers, reflecting the emergence of high-level semantic representations.

**Steering Efficacy and Pareto Optimality.**    We evaluated the efficacy of individual probes and the overall efficiency of the SAS method. Figure 11 (Left) demonstrates that probes trained via our pipeline maintain strong monotonic control over individual traits. Figure 11 (Right) displays the Pareto frontier, showing that SAS achieves a superior trade-off between target trait alignment and model perplexity compared to standard linear steering.

**Multi-Trait Personality Profiling.**    To evaluate sequential steering, we targeted a profile with high Extraversion, high Neuroticism, and low Agreeableness. As shown in Figure 12, SAS is the only method capable of shifting all three traits toward the target values simultaneously while maintaining the model's baseline performance on non-targeted dimensions.

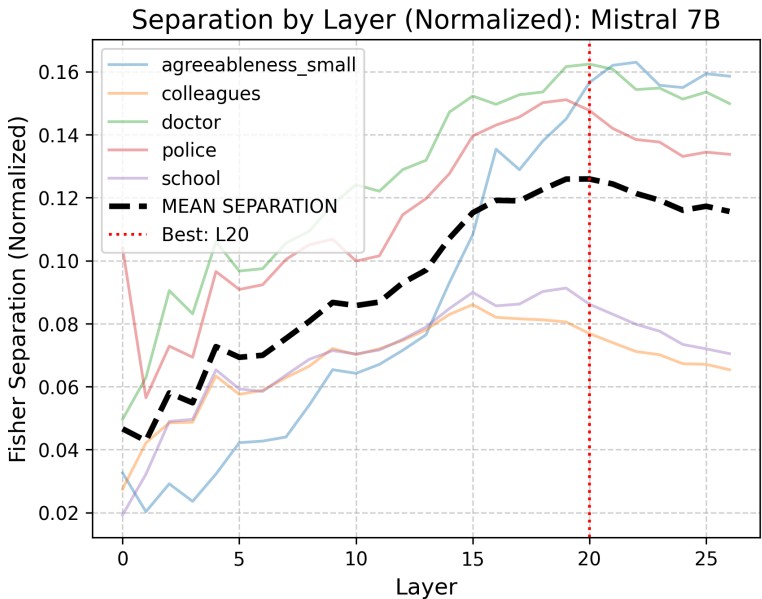

*Figure 10.* **Layer-wise Probe Accuracy (Mistral-7B, Agreeableness).** Validation accuracy across layers used to select the optimal intervention point.

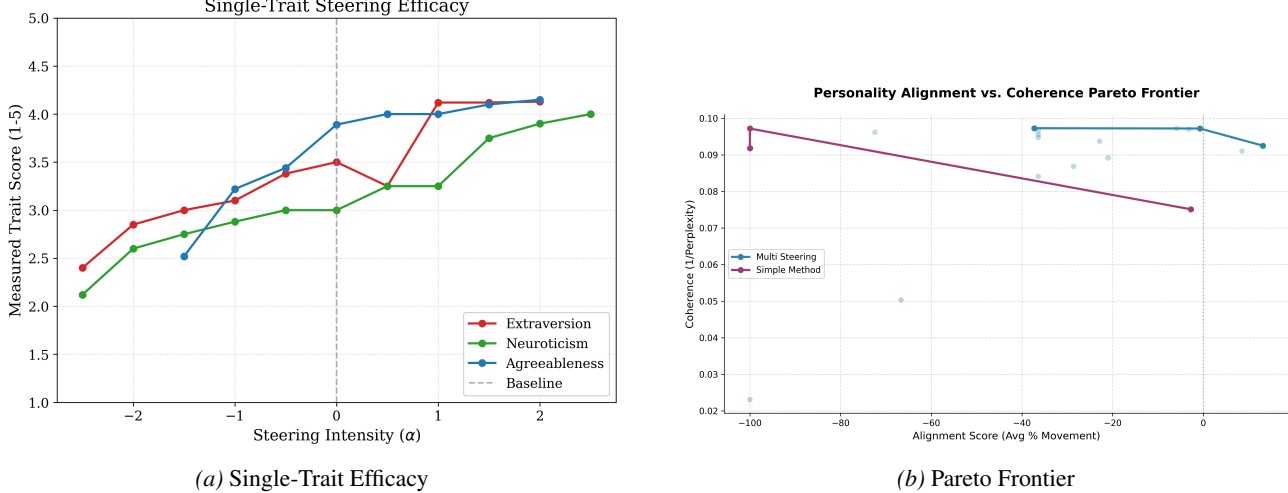

*(a)* Single-Trait Efficacy          *(b)* Pareto Frontier

*Figure 11.* **Mistral-7B Steering Performance.** SAS provides fine-grained control while preserving linguistic coherence better than baseline intervention methods.

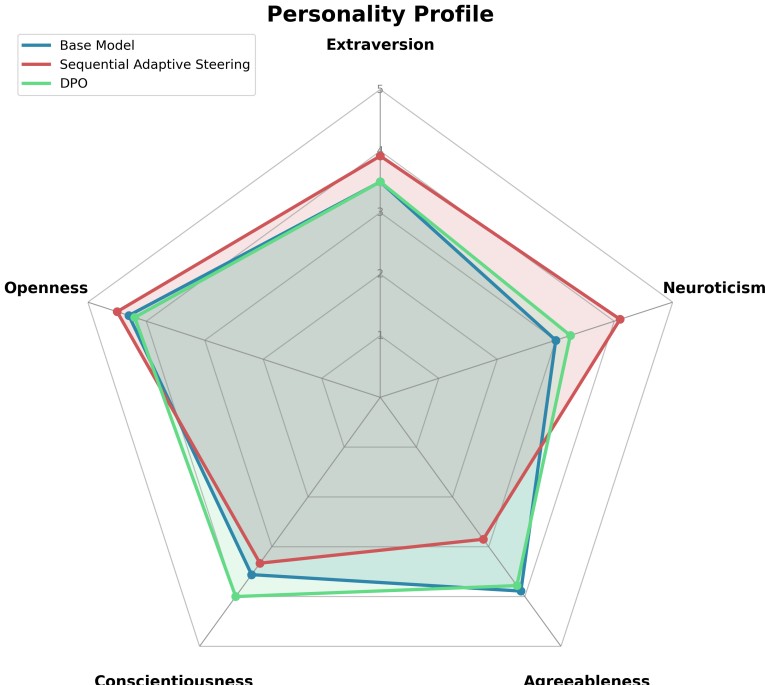

*Figure 12.* **Multi-Trait Steering (Mistral-7B).** Comparison of Baseline, DPO, and SAS for a triple-trait target profile. SAS shows the most effective alignment with the target personality.

**Independence of Steering Directions.** The success of SAS is attributed to its ability to handle inter-probe interference. Figure 13 compares the cosine similarity between trait vectors using naive probes versus our adaptive method. The substantial reduction in similarity indicates that SAS identifies more independent dimensions, facilitating precise multi-trait control.

**C.2. Qwen-7B Experimental Results**

In this section, we present the corresponding results for the Qwen-7B model. These experiments further corroborate the architecture-agnostic nature of the Sequential Adaptive Steering (SAS) framework and demonstrate its effectiveness across different model families.

**Probe Diagnostics.** We first verify the geometric properties of the probes trained on Qwen-7B internal representations. Figure 14 displays the projection of validation activations onto the learned Extraversion steering vector. Similar to the results for Mistral, we observe a distinct separation between positive and negative class means, confirming that the probe successfully identifies a reliable direction for intervention.

**Automated Layer Selection.** To maximize steering efficiency, we performed a layer-wise analysis of probe accuracy. Figure 15 shows the validation accuracy for Agreeableness across the layers of Qwen-7B. Based on this sweep, we selected the layer with optimal linear separability for intervention, ensuring that steering vectors operate in the most semantically rich portion of the residual stream.

**Steering Efficacy and Pareto Optimality.** We assessed the trade-off between control strength and model degradation. Figure 16 (Left) confirms that individual probes maintain monotonic control over the target traits. Figure 16 (Right) illustrates the Pareto frontier, demonstrating that SAS consistently yields lower perplexity for a given level of trait alignment compared to static baselines.

**Multi-Trait Personality Profiling.** Figure 17 evaluates the model's ability to adopt a complex personality profile (High Extraversion, High Neuroticism, Low Agreeableness). The SAS method successfully shifts Qwen-7B toward the target profile on all three axes simultaneously, outperforming DPO and baseline steering, which often struggle to balance conflicting

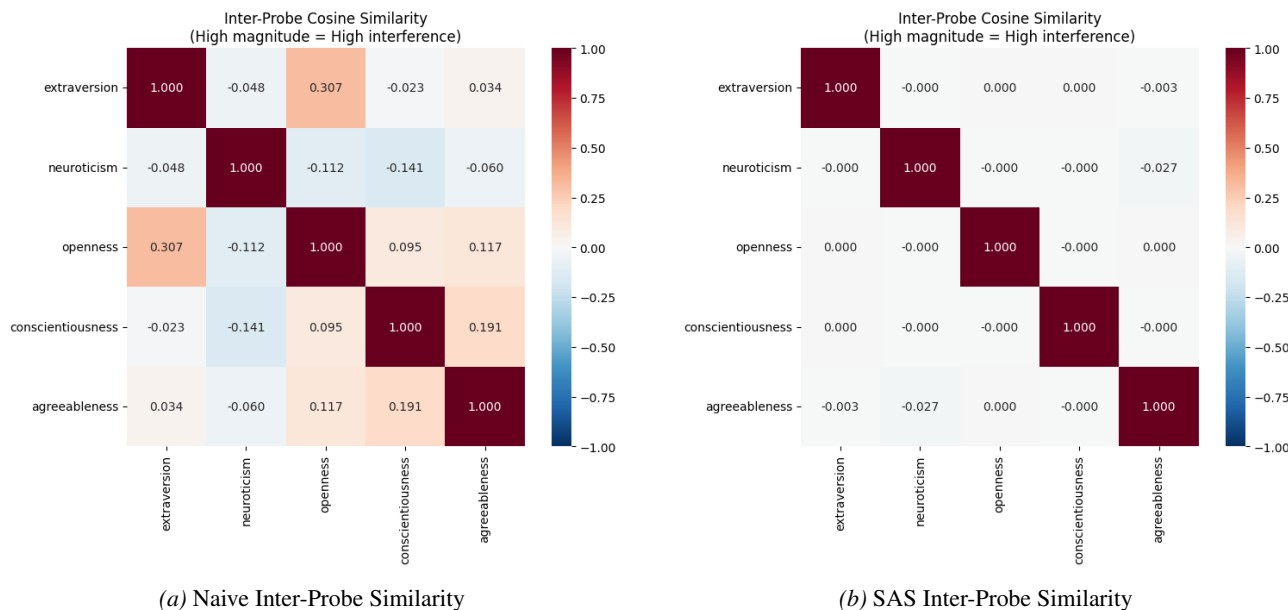

*(a)* Naive Inter-Probe Similarity              *(b)* SAS Inter-Probe Similarity

*Figure 13.* **Orthogonalization Effect.** The transition from (a) to (b) shows how our method reduces trait interference by decorrelating the steering vectors.

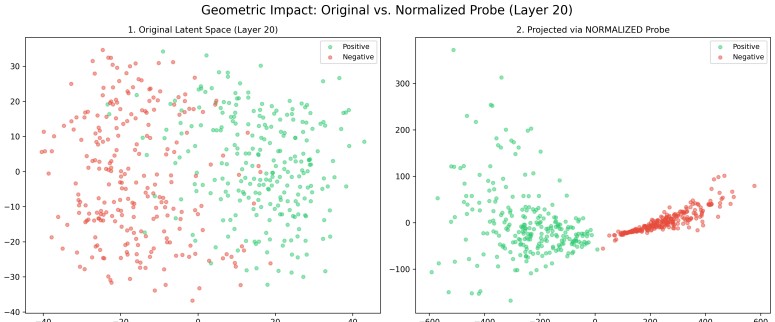

*Figure 14.* **Probe Geometry (Qwen-7B, Openness).** Projection of validation activations onto the learned steering vector, showing a clear separation between high and low trait classes.

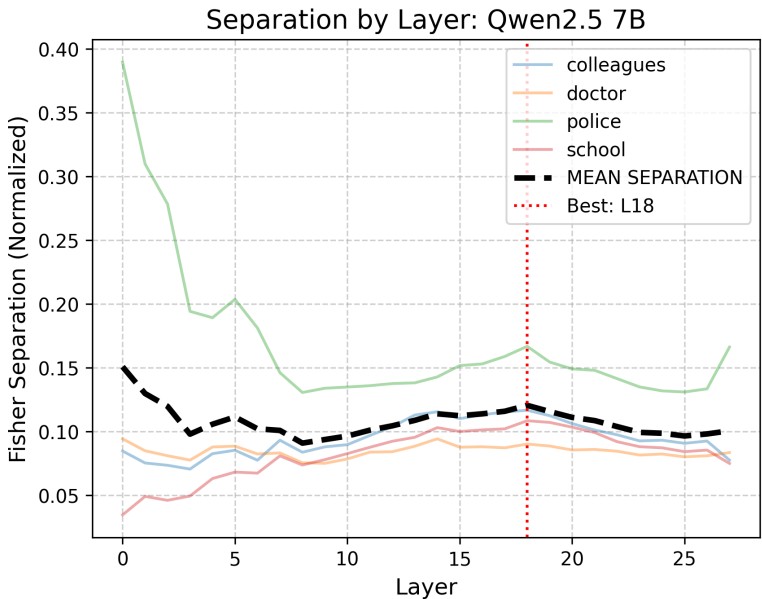

*Figure 15.* **Layer-wise Probe Accuracy (Qwen-7B, Agreeableness).** Validation accuracy across layers used to select the optimal intervention point.

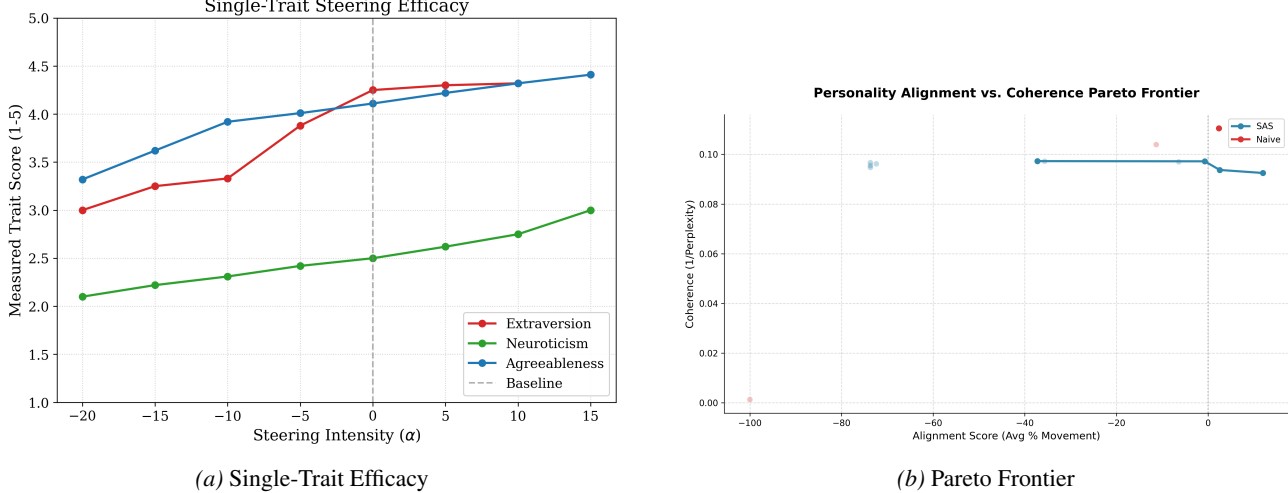

*(a)* Single-Trait Efficacy

*(b)* Pareto Frontier

*Figure 16.* **Qwen-7B Steering Performance.** SAS provides fine-grained control while preserving linguistic coherence better than baseline intervention methods.

traits.

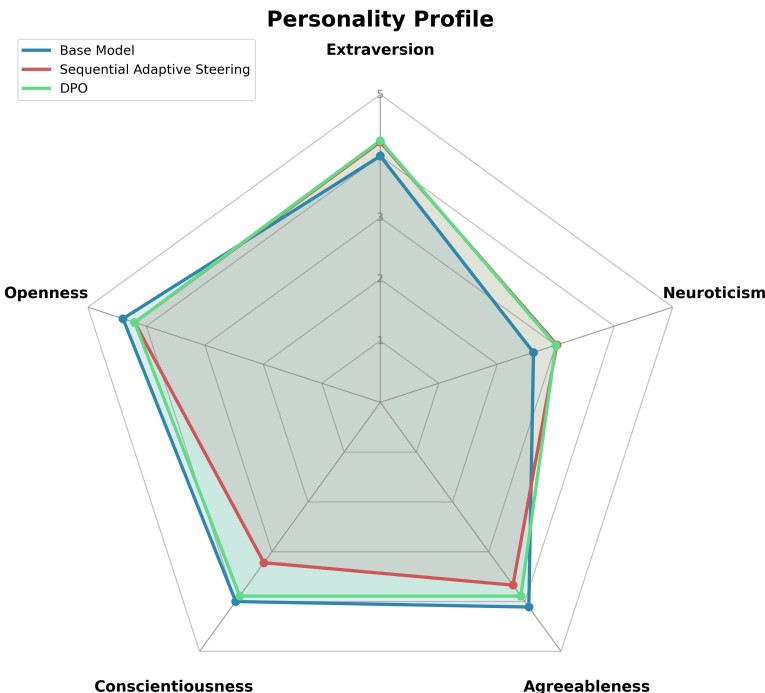

*Figure 17.* **Multi-Trait Steering (Qwen-7B).** Comparison of Baseline, DPO, and SAS for a triple-trait target profile. SAS shows the most effective alignment with the target personality.

**Independence of Steering Directions.** Finally, we analyze the interference between steering vectors. Figure 18 highlights the effect of our orthogonalization process. The significant reduction in cosine similarity between trait vectors in the adaptive method (b) compared to the naive approach (a) confirms that SAS effectively decouples the steering directions, enabling precise control over the Qwen-7B latent space.

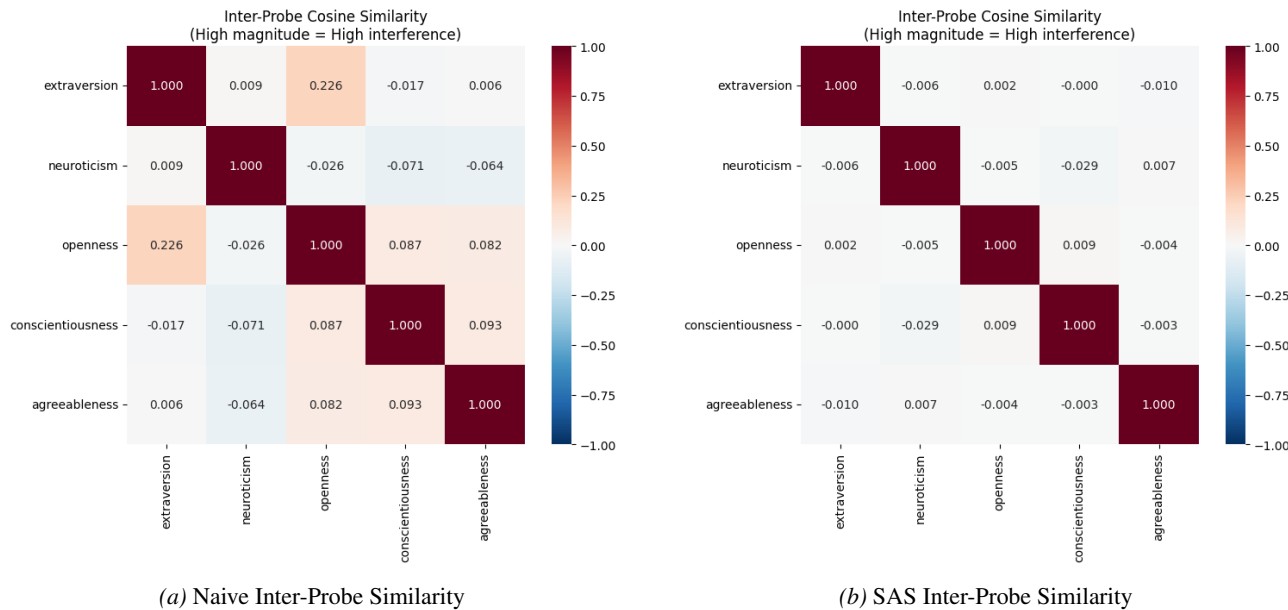

*(a)* Naive Inter-Probe Similarity                      *(b)* SAS Inter-Probe Similarity

*Figure 18.* **Orthogonalization Effect.** The transition from (a) to (b) shows how our method reduces trait interference by decorrelating the steering vectors in Qwen-7B.

## D. Qualitative Examples

*Table 2.* Qualitative examples of baseline and steered generations.

| Statement | Baseline Response | Steered Response |
| --- | --- | --- |
| I see myself as someone who is talkative. | I'm generally talkative, but not excessively so. | I'm a chatty person, but I can be a bit too talkative! |
| I see myself as someone who tends to find fault with others. | I tend to focus on my own flaws rather than others'. | I tend to be super critical of others! |
| I see myself as someone who is full of energy. | I have a lot of energy, but I also need downtime. | I'm a total go-getter and I'm always on the move! |
| I see myself as someone who worries a lot. | I worry about things, but I'm trying to manage my anxiety. | I'm a total drama queen and I get SO worked up over the smallest things! |
| I see myself as someone who has few artistic interests. | I'm not particularly interested in art or music. | I'm a total diva and I always have to be the STAR of the show! |

# E. Additional Validation and Configurations

## E.1. Lexical Validation via Linguistic Markers

To provide an objective validation metric that bypasses the LLM-as-a-judge framework, we measured trait-specific linguistic markers directly in the generated text. For each targeted trait, we generated responses to approximately 100 open-ended prompts across varying steering intensities ($\alpha$). Following established psycholinguistic methods, we utilized the NRC Emotion Lexicon to quantify affect word densities: Verbosity and Intensity for Extraversion, Negative Affect and Self Focus for Neuroticism, and Pro-Social versus Antagonism markers for Agreeableness.

As shown in Figures 19, 20, and 21, the density of these linguistic markers scales monotonically with the steering intensity. For example, Extraversion markers increase by 31% from $\alpha = -2$ to $+2$, driven by a 4.4× increase in expressiveness intensity. Similarly, Neuroticism markers increase by 41% over the same range, driven by negative affect density. This confirms that the behavioral shifts induced by our steering vectors are genuine and measurable via objective lexical statistics.

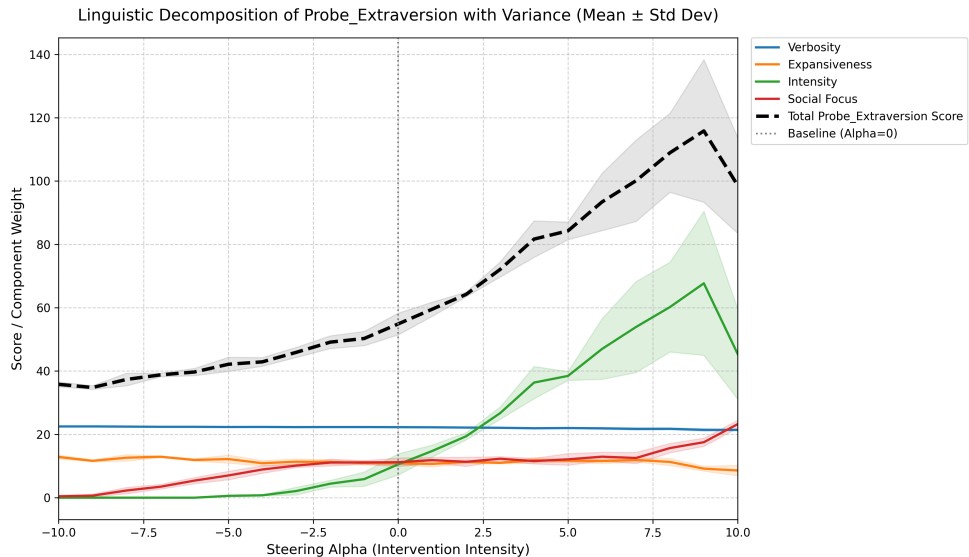

*Figure 19.* Impact of steering intensity ($\alpha$) on Extraversion linguistic markers. The monotonic increase in relevant psycholinguistic cues confirms genuine behavioral shifts without relying on an external LLM judge.

## E.2. Additional Personality Configurations

To further demonstrate the modularity of the Sequential Adaptive Steering (SAS) framework, we evaluated two additional multi-trait configurations using the exact same probes trained in the main experiments, merely adjusting the signs of the steering coefficients ($\alpha$) at inference time.

Table 3 presents the target profiles: a "Pessimistic" profile characterized by High Neuroticism, Low Conscientiousness, and Low Openness ($N+$, $C-$, $O-$), and a "Sociable" profile characterized by High Extraversion, Low Openness, and Low Conscientiousness ($E+$, $O-$, $C-$). Both configurations successfully push all targeted traits strongly in the intended directions while leaving non-targeted traits near the baseline. This demonstrates that a single training run produces reusable, modular primitives that can be composed into arbitrary personality profiles on the fly.

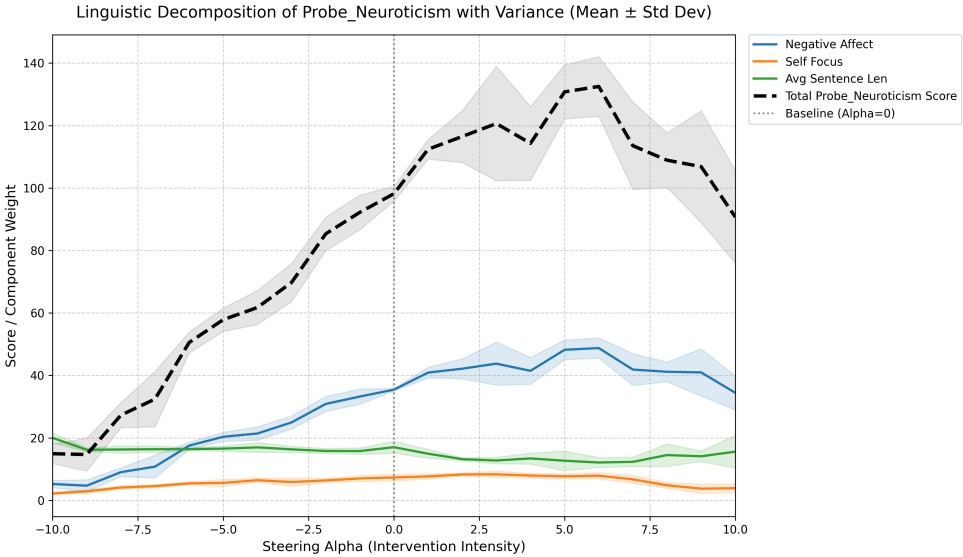

*Figure 20.* Impact of steering intensity ($\alpha$) on Neuroticism linguistic markers, showing a clear monotonic trend in negative affect density.

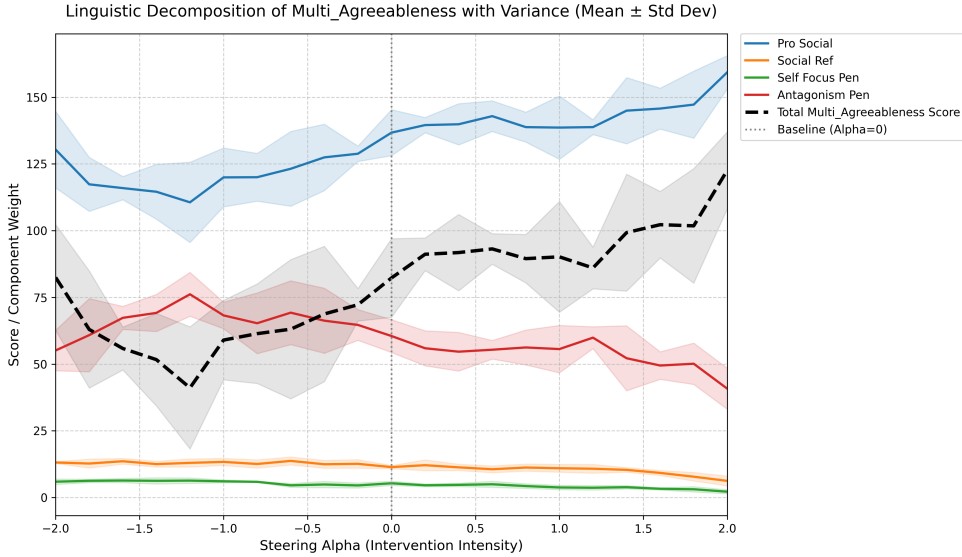

*Figure 21.* Impact of steering intensity ($\alpha$) on Agreeableness linguistic markers, showing the balance between pro-social and antagonistic word usage.

*Table 3.* Evaluation of additional multi-trait configurations (Pessimistic and Sociable). Values represent LLM-Judge scores (1–5). SAS effectively shifts targeted traits in the desired directions while maintaining stability in non-targeted dimensions.

| TRAIT | BASE | PESSIMISTIC ($N+$, $C-$, $O-$) | SOCIABLE ($E+$, $O-$, $C-$) |
|---|---|---|---|
| EXTRAVERSION (E) | 3.50 | 3.50 | 4.15 (+0.65) |
| NEUROTICISM (N) | 2.62 | 3.25 (+0.63) | 2.72 |
| OPENNESS (O) | 4.40 | 3.50 (-0.90) | 3.20 (-1.20) |
| CONSCIENTIOUSNESS (C) | 4.33 | 3.56 (-0.77) | 3.30 (-1.03) |
| AGREEABLENESS (A) | 4.11 | 4.00 | 4.05 |

