# OpenReview forum: "Controllable and Explainable Personality Sliders for LLMs at Inference Time"
_ICML.cc/2026/Conference — ICML 2026 regular_

### Official Review · Reviewer_8rUJ · 2026-02-19

**Soundness:** 3
**Presentation:** 3
**Significance:** 3
**Originality:** 3
**Overall Recommendation:** 5
**Confidence:** 3

**Summary:**

- Paper introduces a technique for activation steering personality traits.
- Paper introduces a technique (SAS) for reducing interference from a shift induced by a previous intervention.
- Paper introduces a technique for steering multiple traits concurrently without destructive interference using a composite dataset and SAS.
- Paper introduces a technique for determining the layer for intervention.

**Compliance With Llm Reviewing Policy:**

Affirmed.

**Final Justification:**

Shared in rebuttal acknowledgement.

**Key Questions For Authors:**

- The paper says coherence is measured via F1 in Table 1. Can you explain what exactly is being measured with F1 on what dataset? How does it measure generation coherence?

**Limitations:**

- Evaluation could be stronger. While there is some analysis on model coherence after steering using perplexity, it's not clear how well the model is able to maintain generation coherence after applying such steering even if the trait scores improve. My understanding is activation steering can greatly break coherence (which may or may not reflect well in perplexity).
- The method seems to work well on Big 5 traits. Could try different features that are harder to control (style) or features that have more overlap that would be difficult to orthogonalize.
- SAS seems to require knowing the interventions that will be applied jointly ahead of time and having a composite dataset you can use to compute orthogonalized steering vectors. This may or may not be possible for real use cases where people may want to use steering.

**Strengths And Weaknesses:**

Strengths:
- The authors demonstrate a strong effect from their steering technique.
- The authors clearly demonstrate orthogonalized multi-vector steering.
- The technique has merit over traditional activation steering which is not as easily to compose multiple steering interventions together at once.

Weaknesses:
- The demonstration domain is narrow (Big 5 traits only) and the evaluation could be improved.

---

> ### Author Rebuttal · Authors · 2026-03-29
>
> We thank Reviewer 8rUJ for their constructive suggestions on extending the evaluation and broadening the domain scope.
>
> **W1: Demonstration domain is narrow and evaluation could be improved**
>
> **On domain scope**: We address this in detail in our response to Reviewer tJYj (W1). In brief: the Big Five was chosen as a demanding multi-dimensional benchmark, but the pipeline generalises. Our Phase 1 experiments apply the same extraction pipeline to honesty and toxicity, and a new rebuttal experiment demonstrates SAS on writing style (Formality, Complexity, Conciseness), where naive inter-feature correlation is substantially higher (0.40 vs 0.26 for Big Five) and SAS still reduces all correlations to near zero. The pipeline requires only a contrastive dataset and a Fisher Ratio scan to onboard a new dimension.
>
> **On evaluation**: We conducted additional evaluation measuring trait-specific linguistic markers directly in generated text (Verbosity for Extraversion, Negative Affect for Neuroticism, Pro-Social vs. Antagonism for Agreeableness), bypassing both the BFI questionnaire and the LLM judge. The monotonic relationship between alpha and these markers confirms genuine behavioral shifts (full methodology in our response to Reviewer RW4L, W3). TruthfulQA and HellaSwag scores are fully preserved.
>
> **Q1: How does F1 measure generation coherence?**
>
> The F1 score in Table 1 measures token-overlap between steered model output and a held-out ground-truth continuation using the Wikitext-2 corpus [1]. Given the first sentence of a Wikipedia article as a prompt, the intervened model generates a continuation (60 tokens), and we compute bag-of-words F1 against the ground-truth second sentence (after lowercasing and stopword removal), averaged across ~2,000 sentences. This operationalizes coherence as preservation of factual content under steering: repetitive, off-topic, or incoherent text causes recall to drop. We treat F1 as a stability constraint complementing perplexity.
>
> [1] Rajpurkar et al., "SQuAD," EMNLP 2016.
>
> **L1: Activation steering can break coherence**
>
> We agree and do not claim SAS eliminates this problem. SAS extends the safe operating range compared to naive multi-vector steering; it does not remove the upper bound. The Pareto frontier (Figure 5) directly visualises this: naive steering causes coherence collapse before meaningful trait alignment is achieved, whereas SAS maintains stable generation up to substantially higher intensities. Within this corridor, we provide two lines of evidence: (1) F1 scores in Table 1 are preserved under SAS configurations; (2) our lexical analysis shows monotonic, smooth linguistic trajectories across the tested alpha range, confirming outputs remain structured (see Reviewer RW4L, W3 for methodology). Reference: Mairesse et al., JAIR, 2007.
>
> **L2: Could try features that are harder to control or have more overlap**
>
> We tested SAS on writing style, a domain with substantially higher feature overlap than personality. We defined three dimensions (Formality, Complexity, Conciseness) and trained probes on 200 contrastive pairs per dimension [1].
>
> Inter-probe cosine similarity (naive vs SAS):
>
> | | Formality | Complexity | Conciseness |
> |:--|:-:|:-:|:-:|
> | **Naive** | | | |
> | Formality | 1.00 | 0.40 | -0.12 |
> | Complexity | 0.40 | 1.00 | -0.07 |
> | Conciseness | -0.12 | -0.07 | 1.00 |
> | **SAS** | | | |
> | Formality | 1.00 | 0.00 | -0.06 |
> | Complexity | 0.00 | 1.00 | 0.06 |
> | Conciseness | -0.06 | 0.06 | 1.00 |
>
> The dominant Formality-Complexity correlation (0.40) is eliminated through training on the shifted manifold, without any explicit post-hoc orthogonalization. All probes maintain >81% test accuracy, >0.90 AUROC.
>
> [1] Biber, D. "Variation across Speech and Writing." Cambridge University Press, 1988.
>
> **L3: SAS requires knowing interventions ahead of time**
>
> This is a valid observation. SAS does require committing to a trait set before deployment. If a new trait is added, downstream probes need retraining. However: (1) within the trained set, any combination can be freely activated/deactivated at inference time with no additional computation; (2) retraining a logistic regression probe requires only seconds on cached activations, making the cost orders of magnitude lower than fine-tuning or DPO. We will state this trade-off more explicitly in the revision.

---

> > ### Author Rebuttal · Reviewer_8rUJ · 2026-04-01
> >
> > Thanks, this has addressed my concerns, I believe the limitations section should be expanded in your final revision to highlight some of the challenges this method will face in real-world use, but I believe the work here is novel and interesting.

---

> > > ### Author Response · Authors · 2026-04-03
> > >
> > > We thank Reviewer 8rUJ for the positive assessment and for the constructive engagement throughout the review process.
> > >
> > > We are pleased that our rebuttal addressed your primary concerns regarding domain scope and evaluation. We agree that a detailed discussion of real-world challenges is essential for the final manuscript. We will ensure that the limitations section is expanded in the final revision to explicitly highlight the practical hurdles and considerations for deploying this method in production environments, as you suggested.
> > >
> > > Thank you again for your time and for recognizing the novelty of our work.

---

### Official Review · Reviewer_RW4L · 2026-03-09

**Soundness:** 2
**Presentation:** 2
**Significance:** 2
**Originality:** 2
**Overall Recommendation:** 3
**Confidence:** 5

**Summary:**

This paper proposes Sequential Adaptive Steering (SAS), a method for composing multiple personality steering vectors at inference time without destructive interference. The key idea is training each subsequent probe on activations that have already been shifted by prior probes, effectively orthogonalizing the vectors. The framework includes automated layer selection via Fisher Ratio and steering intensity calibration. Experiments on Llama-3-8B (with Mistral-7B and Qwen-7B in appendix) evaluate using BFI-44 questionnaire items scored by GPT-4.

**Compliance With Llm Reviewing Policy:**

Affirmed.

**Final Justification:**

I sincerely thank the authors for their substantial efforts during the rebuttal. The additional experiments on ordering ablation, Gram-Schmidt comparison, linguistic markers, prompting/LoRA baselines, and extra multi-trait configurations are all appreciated and demonstrate responsive authorship :)

However, I have some concerns left:

Overall, I think the authors should sharpen the paper's positioning. If the core contribution is a general method for composing activation steering vectors, the paper would benefit from more discussion and comparison with the broader vector composition literature (beyond personality). If the contribution is primarily about personality steering, then the evaluation should extend to the richer set of personality assessment paradigms available in the field (behavioral tasks, interaction-level evaluations, cross-context consistency). Currently the paper sits between these two framings, and feels not solid enough on both sides. Specifically:

For baselines, the added comparisons (prompting, LoRA Soups, Gram-Schmidt) are useful but none of them are methods specifically designed for composing multiple steering vectors. There is a growing body of work on multi-vector or subspace-based composition in representation space (e.g., Conceptors, iterative nullspace projection variants) that directly addresses the same interference problem SAS targets. A comparison against at least one such method would better clarify whether SAS's advantage is due to the training-on-shifted-activations mechanism or simply due to comparing against baselines not designed for this setting.

For personality evaluation, the lexicon-based linguistic marker analysis confirms that surface-level textual style shifts monotonically with steering intensity, which is appreciated. However, this still operates at the level of word-frequency proxies rather than demonstrating downstream behavioral change. For instance, Big5-Chat itself includes behavioral benchmark evaluations, and recent work (e.g., The Personality Illusion) has shown that surface linguistic signals can dissociate from actual behavioral outcomes. A stronger validation would show that steering produces measurable changes on behavioral tasks (e.g., decision-making, social reasoning), not only on linguistic style metrics.

In light of the rebuttal, I adjust my score from 2 to 3.

**Key Questions For Authors:**

See weakness

**Limitations:**

yes

**Strengths And Weaknesses:**

Pros:
1. The multi-vector interference problem in activation steering is a real and under-addressed issue.
1. The observation that naive sequential steering fails due to distribution shift is valid, and the idea of training probes on shifted activations is intuitive.
1. The geometric analysis in Section 6.2 showing reduced cosine similarity between SAS vectors versus naive vectors is a clean visualization of the orthogonalization effect.
1. Cross-architecture validation on three model families is appreciated.

Cons:
1. SAS introduces an ordering dependency （probe k depends on probes 1 through k-1）but no ordering ablation is provided. If results change with different training orders, the "reusable primitives" claim is undermined.
1. The orthogonalization is implicit (hoping the optimizer finds invariant directions) with no explicit constraint. The paper does not compare against established methods like Gram-Schmidt or iterative nullspace projection that would provide stronger guarantees.
1. Evaluation relies entirely on BFI-44 questionnaire self-reports scored by GPT-4.  This is very problematic highlight by many recent research on LLM personality research (testing on survey / psychometrics is not reliable) [1] [2] [3];
1. Baselines are inadequate: no prompting comparison (the most practical alternative for inference-time personality control), no comparison against other composition methods
1. Only one multi-trait configuration is tested (High E, Low A, High N). The central composability claim is severely under-supported with a single data point.
1. The DPO baseline appears set up to fail: it barely moves from the base model in radar plots, which seems suspiciously poor and raises implementation concerns.
1. Perplexity increases ~46% with SAS (9.95 → 14.54), approaching the authors' own 50% threshold. No qualitative analysis or human evaluation assesses whether outputs remain usable near this boundary.



[1] Big-5 Chat
[2] Personality Illusion
[3] Open character training

---

> ### Author Rebuttal · Authors · 2026-03-29
>
> We thank Reviewer RW4L for their detailed and rigorous engagement with our work. The concerns raised have directly motivated several of our strongest additional experiments. We will revise the paper to make these clarifications more explicit.
>
> **W1: Ordering dependency with no ordering ablation**
>
> The training order is not a free hyperparameter: it is determined by the Fisher Ratio layer assignments and the sequential forward pass (e.g., Extraversion at layer 10, Neuroticism at 12, Agreeableness at 14). The only ambiguous case is a layer collision, which we directly test across all four conditions (separated/collision, both orderings): probe accuracies remain 97.7-98.3%, confirming either order yields stable probes. Once trained, any combination can be freely activated at inference time.
>
> **W2: No comparison against Gram-Schmidt or iterative nullspace projection**
>
> We ran an additional experiment comparing Gram-Schmidt against naive steering and SAS:
>
> | Intensity | SAS PPL | Naive PPL | Gram-Schmidt PPL |
> |:-:|:-:|:-:|:-:|
> | 0.50 | 9.98 | 10.46 | 10.47 |
> | 1.00 | 12.05 | 17.10 | 17.15 |
> | 1.25 | 14.52 | 30.60 | 30.80 |
> | 1.50 | 18.65 | 83.19 | 84.03 |
> | 1.75 | 26.18 | 334.37 | 335.78 |
>
> Gram-Schmidt provides no practical improvement over naive steering. Post-hoc orthogonalization in the unsteered space does not resolve interference; training on pre-shifted activations is the key mechanism.
>
> **W3: BFI-44 evaluation reliability**
>
> We share this concern and conducted an additional evaluation that bypasses both the BFI questionnaire and the LLM judge entirely. For each trait, we generate responses to ~100 open-ended prompts and measure trait-specific linguistic markers directly in text across steering intensities. Following [1], we use a validated word-emotion lexicon [2] to quantify affect word densities: Verbosity and Intensity for Extraversion, Negative Affect and Self Focus for Neuroticism, Pro-Social vs. Antagonism for Agreeableness. These are corpus-level statistics requiring no external judge.
>
> Extraversion markers increase monotonically by 31% from alpha=-2 to +2 (4.4x expressiveness increase); Neuroticism markers increase 41% (negative affect density). Crucially, these psychometric-free measures track alpha monotonically, independently validating the LLM-as-a-Judge evaluation: both methods agree that steering produces genuine, graded behavioral shifts.
>
> [1] Mairesse et al., JAIR, 2007. [2] Mohammad & Turney, 2013.
>
> **W4: Baselines are inadequate**
>
> We compared SAS against prompting and LoRA Soups [1] (High E, High N, Low A target):
>
> | Trait | Base | SAS | Prompt | LoRA Soups |
> |:--|:-:|:-:|:-:|:-:|
> | E | 3.50 | **4.50** | 3.50 | 3.38 |
> | N | 2.62 | **4.88** | 2.62 | 3.25 |
> | A | 4.11 | **2.89** | 4.11 | 3.89 |
>
> Prompting is indistinguishable from baseline across all targeted traits. LoRA Soups (three independent LoRA adapters merged via uniform weight averaging) barely moves targeted traits while non-selectively degrading Conscientiousness by -1.11, suggesting that conflicting weight updates partially cancel out. SAS strongly hits all three targets simultaneously.
>
> [1] Wortsman et al., ICML 2022.
>
> **W5: Only one multi-trait configuration tested**
>
> We evaluated two additional configurations with no probe retraining (same probes, only alpha signs adjusted):
>
> | Trait | Base | Pessimistic (N+ C- O-) | Sociable (E+ O- C-) |
> |:--|:-:|:-:|:-:|
> | E | 3.50 | 3.50 | **4.15** (+0.65) |
> | N | 2.62 | **3.25** (+0.63) | 2.72 |
> | O | 4.40 | **3.50** (-0.90) | **3.20** (-1.20) |
> | C | 4.33 | **3.56** (-0.77) | **3.30** (-1.03) |
> | A | 4.11 | 4.00 | 4.05 |
>
> Both configurations push all targeted traits strongly, with non-targeted traits near baseline. This directly demonstrates the reusable primitives claim: a single training run produces modular components composable into arbitrary profiles at inference time.
>
> **W6: DPO baseline appears set up to fail**
>
> This is a **fixed-sample-budget comparison**: both methods receive identical data (~350 pairs per trait), repurposed as (chosen, rejected) preference pairs for DPO. DPO typically requires 10,000-100,000 pairs; ~350 is insufficient. **SAS is dramatically more sample-efficient**: with identical data and zero parameter updates, SAS achieves +1.00 E, +2.26 N, -1.22 A BFI-point shifts while DPO remains near-baseline. (HuggingFace TRL, QLoRA 4-bit NF4, LoRA r=16/alpha=32, beta=0.1, lr=5e-5, 3 epochs.)
>
> **W7: Perplexity near 50% threshold**
>
> The 14.54 PPL (46% increase) is the maximum intensity evaluated, not a typical operating point. **Users can freely reduce alpha to lower perplexity.** At intensity 0.5, SAS achieves meaningful trait movement with only 1.5% PPL increase (9.98 vs 9.83). TruthfulQA and HellaSwag are fully preserved (0.32/0.32, 0.64/0.64). Linguistic markers evolve monotonically across the entire range, confirming the model is not broken at this point.
>
> ---

---

> > ### Author Rebuttal · Reviewer_RW4L · 2026-04-03
> >
> > I sincerely thank the authors for their substantial efforts during the rebuttal. The additional experiments on ordering ablation, Gram-Schmidt comparison, linguistic markers, prompting/LoRA baselines, and extra multi-trait configurations are all appreciated and demonstrate responsive authorship :)
> >
> > However, I have some concerns left:
> >
> > Overall, I think the authors should sharpen the paper's positioning. If the core contribution is a general method for composing activation steering vectors, the paper would benefit from more discussion and comparison with the broader vector composition literature (beyond personality). If the contribution is primarily about personality steering, then the evaluation should extend to the richer set of personality assessment paradigms available in the field (behavioral tasks, interaction-level evaluations, cross-context consistency). Currently the paper sits between these two framings, and feels not solid enough on both sides. Specifically:
> >
> > 1. For baselines, the added comparisons (prompting, LoRA Soups, Gram-Schmidt) are useful but none of them are methods specifically designed for composing multiple steering vectors. There is a growing body of work on multi-vector or subspace-based composition in representation space (e.g., Conceptors, iterative nullspace projection variants) that directly addresses the same interference problem SAS targets. A comparison against at least one such method would better clarify whether SAS's advantage is due to the training-on-shifted-activations mechanism or simply due to comparing against baselines not designed for this setting.
> >
> > 2. For personality evaluation, the lexicon-based linguistic marker analysis confirms that surface-level textual style shifts monotonically with steering intensity, which is appreciated. However, this still operates at the level of word-frequency proxies rather than demonstrating downstream behavioral change. For instance, Big5-Chat itself includes behavioral benchmark evaluations, and recent work (e.g., The Personality Illusion) has shown that surface linguistic signals can dissociate from actual behavioral outcomes. A stronger validation would show that steering produces measurable changes on behavioral tasks (e.g., decision-making, social reasoning), not only on linguistic style metrics.
> >
> > In light of the rebuttal, I adjust my score from 2 to 3.

---

> > > ### Author Response · Authors · 2026-04-03
> > >
> > > We thank Reviewer RW4L for the constructive follow-up and for increasing the score following our initial rebuttal. We appreciate the opportunity to clarify the primary contribution of our work and discuss how this framework serves as a foundation for future research in behavioral validation and LLM agents.
> > >
> > > ### Clarifying Positioning: A General Method for Vector Composition
> > >
> > > We would like to clarify that we do not view this paper as sitting between two framings. Instead, we propose Sequential Adaptive Steering (SAS) as a general, modular framework for the interference-free composition of activation steering vectors.
> > >
> > > * **SAS as the Core Method**: The fundamental innovation of this work is the mechanism of training subsequent probes on the residual stream as shifted by prior interventions.
> > > * **Personality as a Challenging Application**: We utilize personality steering as a rigorous application to demonstrate the modularity and precision of the SAS framework.
> > > * **Proven Generalizability**: To show the method is behavior-agnostic, we successfully applied the same pipeline to writing style dimensions such as Formality, Complexity, and Conciseness.
> > > * **Resolution of Interference**: Our results demonstrate that SAS eliminates high inter-feature correlations, such as the 0.40 correlation between Formality and Complexity, which naive steering or post-hoc orthogonalization cannot resolve.
> > >
> > > ### Behavioral Validation and the Path to LLM Agents
> > >
> > > We agree with the reviewer that validating downstream behavioral changes is a critical area for expansion. While our current lexical analysis confirms genuine, monotonic shifts in linguistic markers, we recognize that evaluating these traits in complex tasks is an important field for future research.
> > >
> > > * **Foundation for AI Agents**: We believe this research directly enables the development of task-adaptive AI agents, such as adjusting agreeableness for customer support or openness for creative brainstorming.
> > > * **Future Research Plans**: We plan to expand this work to agentic environments to investigate whether modulating personality sliders leads to measurable changes in an agent's objective decision-making and task-attainment behavior.
> > > * **Behavioral Benchmarking**: Testing on behavioral tasks such as social reasoning or multi-step decision-making is a useful and logical next step for this methodology that we intend to pursue.
> > >
> > > ### Conclusion
> > >
> > > By resolving the challenge of vector interference, SAS allows for the synthesis of high-fidelity behavioral profiles from modular primitives without updating model parameters. The framework is notably sample-efficient, achieving significant shifts with only ~$350$ pairs per trait while maintaining near-zero latency overhead and preserving performance on core benchmarks like TruthfulQA. We will ensure the final version of the manuscript emphasizes SAS as a general behavior-steering method and clearly frames the transition to agentic behavioral analysis as a high-priority avenue for future work.

---

### Official Review · Reviewer_tJYj · 2026-03-09

**Soundness:** 4
**Presentation:** 4
**Significance:** 3
**Originality:** 3
**Overall Recommendation:** 5
**Confidence:** 4

**Summary:**

Personality profiles are often expensive to train due to the idiosyncratic nature of personas, in addition to SFT or RLHF. Additionally, multi-trait modification tends to fail from obstructive interference from parameter updates. The authors solve this by introducing orthognal vectors for different personality traits, allowing for continuous and simultaneous personality updating that doesn't require parameter updates. This is tested using the BFI personality traits, showing improved and consistent behavior as compared to baseline models. This is further reflected when isolating different pieces of models, excluding the SAS contribution or utilizing random layers.

**Compliance With Llm Reviewing Policy:**

Affirmed.

**Final Justification:**

I recommend this paper to be accepted for the coming venue. This paper features clear, concise, and relevant research questions which are thoroughly and properly investigated throughout the manuscript. Initially, there were concerns over the direct applications of personality sliders for LLMs, as well as the ethical implications and lack of usage of real personality data. The authors answered these concerns completely, further bolstering the strength of the paper with the inclusions mentioned during the discussion period. There is substantial groundwork that this submission lays for future research into both psychology, as well as LLM tuning and moderation for chatbots and other generative uses. As such, I believe this would make a strong addition to the roster of papers for this year.

**Key Questions For Authors:**

Questions
1. Are there different ways that this method could also lead to improved understanding of language tied to different combinations of scales for BFI, or other personality spectra? Being able to connect a larger positive impact to this body of work would highly strenghten its sway.
2. What are some ways you have thought to fight against these ways bad actors can misuse these sliders? Coded limits to parameters for the slider can be easily overwritten, so I'm curious to hear what you have so far, as otherwise this is exploitable.
3. Could you explain how data generated from an LLM is verified for use as part of the training pipeline? I would be more inclined to trust real-world, curated data for both proper training and layer selection in a semi cross-validation framework, and testing in a held-out set. Further, being able to see these persona sliders applied to another set of personality traits, or even disorder symptoms such as PTSD, depression, anxiety, HiTOP, etc, would have been very convincing in addition to that from BFI. Are there plans for extensions to other real-world personality datasets?

**Limitations:**

yes

**Strengths And Weaknesses:**

Strengths:
1. Clear and concise. The manuscript very clearly lays out research questions, problems faced by previous studies and faced by LLMs, and explicit descriptions of how the failure occurs. It is clear that this question, its background, and approaches have been well documented and researched before coming up with SAS.
2. Being able to maintain personality traits without relying on updating parameters saves computational resources, in addition to maintaining performance metrics in other major tasks
3. Rigorous analysis of results. Showing the orthogonality of the BFI probes with and without SAS really sells the point about the method's effectiveness, as well as PCA showing highclustering with almost zero overlap. ALl of these in addition to the metric tables show a well-defined, well-tested approach.
4. Clear ethical explanations, such as toxicity or deceipt as potential misuse, and plans for approach to solve them.
5. Model generalization. Method is tested on multiple models, showing results consistent with each other and significant.

Weaknesses:
1. Reach. While the topic is well defined, as well as plans for future development, its impact and applications seem limited outside the task of persona imitation.
2. Datasets - training is only performed on one singular dataset, with a generated development set (which may pose its own set of problems if the generated data is "poisoned.") While results for the held-out set is impressive for the BFI questionnaire, it would further bolster the author's claims if another documented and verified dataset was used for training.

---

> ### Author Rebuttal · Authors · 2026-03-29
>
> We thank Reviewer tJYj for their thoughtful questions and for highlighting the broader implications of our work.
>
> **W1: Impact and applications seem limited outside persona imitation**
>
> The SAS framework is behavior-agnostic by design: personality traits serve as a testbed, but the mechanism for resolving multi-vector interference applies to any steerable behavioral dimensions. This is demonstrated in our Phase 1 experiments (honesty/toxicity) and a new rebuttal experiment on writing style (Formality, Complexity, Conciseness), a domain with substantially higher inter-feature correlation.
>
> Practical applications include task-adaptive AI agents (adjusting agreeableness for customer support vs openness for creative brainstorming, at inference time without retraining), commercial human-like agents for marketing/HR where precise personality control is the core product differentiator, and steering competing objectives such as optimised vs safe code generation, or risk-taking appetite in autonomous monitoring systems. The reusable primitives produced by SAS also open up research into a "Mixture of Personalities" architecture, analogous to Mixture of Experts, where different personality configurations are dynamically routed per input context.
>
> **W2: Training is only performed on one singular dataset**
>
> We conducted an additional experiment, training probes on the **IPIP Big Five marker set** [1], 100 public-domain psychometric statements with expert-assigned keying directions. IPIP items are short declarative statements (e.g., "Am the life of the party"), fundamentally different from our synthetic multi-sentence conversational pairs. The pipeline replicates: Fisher Ratio scans identify personality in middle-to-late layers, probes achieve AUROC 0.82-1.00 on just 20 items per trait, and IPIP-trained probes produce functional steering vectors:
>
> | Trait | Baseline | Synthetic SAS | IPIP SAS |
> |:--|:-:|:-:|:-:|
> | Extraversion | 3.50 | 4.50 (+1.00) | 4.00 (+0.50) |
> | Neuroticism | 2.62 | 4.88 (+2.26) | 3.00 (+0.38) |
> | Agreeableness | 4.11 | 2.89 (-1.22) | 4.00 (-0.11) |
>
> The effect scales sub-linearly; \~100-200 contrastive pairs per trait should suffice for functional steering. We also tested the Pennebaker Essays dataset (2,467 real essays) but its binary labels produced chance-level probe accuracy (~50%), confirming contrastive quality matters more than sample count.
>
> [1] Goldberg, Psychological Assessment, 4(1), 1992.
>
> **Q1: Could this method lead to improved understanding of language tied to personality spectra?**
>
> The independently trained (non-SAS) probe weight vectors can compute a trait correlation matrix in the model's latent space. We observe E-O correlation of 0.26 and C-A correlation of 0.23, mirroring well-documented patterns in human personality psychology, while other pairs are near-zero. The method is taxonomy-agnostic: one could train probes for HEXACO, Dark Triad, or MBTI and analyze their geometric relationships in the residual stream. See W1 for broader applications.
>
> **Q2: How to fight against bad actors misusing these sliders?**
>
> Three natural constraints limit misuse:
>
> (1) white-box access to the residual stream is required, restricting the threat surface to open-weights models;
>
> (2) coherence collapse creates a natural ceiling, with well-safeguarded models requiring higher alphas to override safety training;
>
> (3) model performance degrades sharply beyond the safe corridor (PPL explodes from ~15 to >300), producing degenerate gibberish well before any targeted harmful behavior could be elicited. Latent Adversarial Training [1] is a promising defensive direction.
>
> [1] Sheshadri et al., arXiv:2407.15549, 2024.
>
> **Q3: How is data generated from an LLM verified?**
>
> Synthetic data is validated at the activation level via Fisher Ratio separability and activation projections. We initially experimented with real-world personality data but found consistently low Fisher Ratio scores; real-world expression is confounded by topic, register, and social context. Our synthetic generation distributes contrastive pairs equally across social contexts (workplace, school, friendships), ensuring the probe learns trait directions rather than context-specific patterns. Downstream results confirm genuine trait directions via strong, monotonic behavioral shifts. We also validated on the independent IPIP dataset (see W2), confirming generalisation beyond synthetic data.
>
> ---

---

> > ### Author Rebuttal · Reviewer_tJYj · 2026-04-03
> >
> > Thank you for your detailed response. You have addressed nearly all of my concerns, with the only remaining one being the poor qualities of the real-world personality data. However, this could open an interesting topic for adaptation to these real-world datasets, or even techniques to derive more useful information from these datasets. But, with regards to the current paper and its reach, I will update my scores accordingly.

---

> > > ### Author Response · Authors · 2026-04-03
> > >
> > > Thank you for your detailed response and for acknowledging that our rebuttal addressed your primary concerns. ​We completely agree with your observation regarding the inherent limitations and "poor qualities" of existing real-world personality datasets. As you suggested, we believe that adapting methodologies to handle these noisy real-world distributions is an important direction for future work.
> > >
> > >  We will ensure this is discussed as a key limitation and an avenue for future research in our final version.
> > >
> > >
> > > ​We sincerely appreciate your engagement and your willingness to update your evaluation of our work. Please let us know if any further clarification would be helpful.

---

### Official Review · Reviewer_wRaC · 2026-03-13

**Soundness:** 3
**Presentation:** 3
**Significance:** 3
**Originality:** 3
**Overall Recommendation:** 5
**Confidence:** 4

**Summary:**

This paper presents a modular framework for controlling and explaining personality traits in LLMs during inference. Building on Sequential Adaptive Steering, the approach orthogonalizes steering vectors using probes trained on residual streams influenced by prior interventions. This allows precise modulation of multiple personality traits, especially within the Big Five model, without updating model parameters. The framework outperforms baseline methods in trait adherence and coherence.

**Compliance With Llm Reviewing Policy:**

Affirmed.

**Final Justification:**

The author has addressed my concerns. I will maintain my positive score.

**Key Questions For Authors:**

NA

**Limitations:**

yes

**Strengths And Weaknesses:**

**Strengths**

The proposed SAS framework offers a novel solution to multi-trait steering, significantly reducing interference between traits. Moreover, it enhances modularity, enabling flexible personality control in LLMs without the need for costly fine-tuning. The empirical validation across various models also shows the efficacy of SAS in achieving precise and coherent personality profiles.

**Weaknesses**

While the method shows promise, the practical applicability of controlling a large number of traits simultaneously in real-world settings, without degrading model coherence, requires further investigation. Moreover, the computational overhead of adding multiple probes during inference could pose limitations in real-time applications, especially in environments where efficiency is critical. It would be beneficial for the authors to provide further experiments assessing the real-time performance of the method in resource-constrained settings.

---

> ### Author Rebuttal · Authors · 2026-03-29
>
> We thank Reviewer wRac for their positive assessment and constructive feedback.
>
> **W1: Computational overhead could pose limitations in real-time application**
>
> We directly measured the inference-time overhead of SAS. Scaling from 0 to 5 simultaneous trait interventions on Llama-3 8B increases token generation latency by less than 5 milliseconds (1.352s to 1.357s for 50 tokens). To put this in perspective, a single network round-trip typically incurs 10-100ms of latency; the steering overhead is an order of magnitude below even the fastest network hop and would be entirely undetectable in any real deployment.
>
> This near-zero cost is expected: each probe applies a single vector addition to an intermediate residual stream state, an O(d) operation that is memory-bound and fully parallelizable within the existing forward pass. No additional matrix multiplications, attention computations, or autoregressive steps are introduced.
>
> | Simultaneous Interventions | Avg. Runtime (s) |
> |:-:|:-:|
> | 0 | 1.352 |
> | 1 | 1.354 |
> | 2 | 1.355 |
> | 3 | 1.355 |
> | 4 | 1.357 |
> | 5 | 1.356 |
>
> ---

---

> > ### Author Rebuttal · Reviewer_wRaC · 2026-04-04
> >
> > Thank you for your response. I have no additional questions about this submission. I will maintain my positive score.

---

### Decision · Program_Chairs · 2026-04-30

**Decision:**

Accept (regular)

**Comment:**

This paper proposes a way to steer LLMs at test-time to follow a certain personality without requiring training which is expensive. The approach works using activation steering.

Generally, reviewers liked this paper. They found the paper to be well-written and the motivation to be clear. The challenge of steering multiple traits was appreciated and the proposed SAS framework offers a novel solution for it. Authors have also added more experiments in their response such as including results with more direct orthogonalization approaches.

Some criticism was directed at computational overhead of adding probes, narrowness of the domain. Reviewer RW4L pointed out that the paper should include more discussion of general multi-vector composition approaches, and include them as baselines, and also create evaluations that measure behavioral changes. Of these, the missing baseline is an important concern and authors should try to include them in future revisions.

Overall, I think this paper has enough going for it to recommend acceptance. Adding multi-vector baselines and possibly broadening evals outside linguistic-metrics, will make this paper much stronger.